
# 1 Spatiotemporal Variability in the Oxidative Potential of Ambient 2 Fine Particulate Matter in Midwestern United States

Haoran Yu[1], Joseph Varghese Puthussery[1], Yixiang Wang[1], Vishal Verma[1]*
[1]Department of Civil and Environmental Engineering, University of Illinois at Urbana-Champaign, Urbana, IL, 61801,
United States
* *Correspondence to:* Vishal Verma (vverma@illinois.edu)
**Abstract.** We assessed the oxidative potential (OP) of both water-soluble and methanol-soluble fractions of ambient
fine particulate matter ($PM_{2.5}$) in the midwestern United States. A large set of $PM_{2.5}$ samples (N = 241) were collected
from five sites, setup in different environments, i.e. urban, rural and roadside, in Illinois, Indiana and Missouri during
May 2018 – May 2019. Five acellular OP endpoints, including the consumption rate of ascorbic acid and glutathione
in a surrogate lung fluid (SLF) ($OP^{AA}$ and $OP^{GSH}$, respectively), dithiothreitol (DTT) depletion rate ($OP^{DTT}$), and ·OH
generation rate in SLF and DTT ($OP^{OH-SLF}$ and $OP^{OH-DTT}$, respectively), were measured for all $PM_{2.5}$ samples. $PM_{2.5}$
mass concentrations in the Midwest US as obtained from these samples were spatially homogeneously distributed,
while most OP endpoints showed significant spatiotemporal heterogeneity. Seasonally, higher activities occurred in
summer for most OP endpoints for both water- and methanol-soluble extracts. Spatially, roadside site showed highest
activities for most OP endpoints in the water-soluble extracts, while only occasional peaks were observed at urban
sites in the methanol-soluble OP. Most OP endpoints showed similar spatiotemporal trends between mass- and
volume-normalized activities across different sites and seasons. Comparisons between two solvents (i.e. water and
methanol) showed that methanol-soluble OP generally had higher activity levels than corresponding water-soluble
OP. Site-to-site comparisons of OP showed stronger correlations for methanol-soluble OP compared to water-soluble
OP, indicating a better extraction of water-insoluble redox-active compounds from various emission sources into
methanol. We found a weak correlation and inconsistent slope values between $PM_{2.5}$ mass and most OP endpoints.
Moreover, the poor-to-moderate intercorrelations among different OP endpoints infer different mechanisms of OP
represented by these endpoints, and thus demonstrate the rationale for analyzing multiple acellular endpoints for a
better and comprehensive assessment of OP.

## 26 1 Introduction

Oxidative stress induced by ambient fine particulate matter ($PM_{2.5}$; particulate matter with size less than 2.5 μm) has
been widely recognized as a biological pathway for fine particles to exert adverse health effect in humans (Sørensen
et al., 2003;Risom et al., 2005;Garçon et al., 2006;Wessels et al., 2010;Cachon et al., 2014;Haberzettl et al., 2016;Feng
et al., 2016;Rao et al., 2018;Mudway et al., 2020). A variety of chemical species in ambient particles, such as transition
metals and aromatic organic species, possess redox cycling capability and can catalyze electron transfer from cellular
reductants (e.g. NADPH) to molecular oxygen ($O_2$), which subsequently forms highly reactive radicals [e.g.
superoxide radical ($·O_2^-$) and hydroxyl radical (·OH)] and non-radical oxidants [e.g. hydrogen peroxide ($H_2O_2$)]





(Kampfrath et al., 2011;Qin et al., 2018;Kumagai et al., 2002;Lee et al., 2016). These oxygen containing species with
high redox activity and short lifetime are collectively defined as the reactive oxygen species (ROS). Several
antioxidants (e.g. ascorbic acid (AA), reduced glutathione (GSH) and uric acid (UA) etc.) that are present in human
respiratory tract lining fluid (RTLF) can counteract the ROS under normal conditions by donating extra electrons, thus
forming less-oxidative species and oxidized antioxidants (Kelly, 2003;Li and Nel, 2006;Allan et al., 2010;Zuo et al.,
2013;Poljšak and Fink, 2014). However, excessively produced ROS might penetrate the antioxidant barrier and induce
oxidative stress (Xing et al., 2016;Rao et al., 2018), leading to the cascade of detrimental biological effects such as
oxidation of DNA, lipids and proteins (Rossner et al., 2008;Franco et al., 2008;Grevendonk et al., 2016), tissue injury
(Feng et al., 2016;Gurgueira et al., 2002;Sun et al., 2020) and eventually cardiopulmonary impairment (Li et al.,
2018;Kodavanti et al., 2000;Kampfrath et al., 2011). The capability of particulate matter (PM) for catalyzing the
generation of ROS and/or the depletion of antioxidants is defined as the oxidative potential (OP) of PM (Bates et al.,

45   2019).

The assessment of $PM_{2.5}$-induced oxidative stress is conventionally carried out through biological tests, including both
*in vitro* (Becker et al., 2005;Zhang et al., 2008;Oh et al., 2011;Yan et al., 2016;Abbas et al., 2016;Deng et al., 2013)
and *in vivo* designs (Kleinman et al., 2005;Riva et al., 2011;Pei et al., 2016;Araujo et al., 2008;Xu et al., 2011;Sancini
et al., 2014). Although, these biological tests are highly relevant in terms of representing the health effects in humans,
the time- and labor-intensive protocols as well as the cost of experimental materials generally limit their application
to only small sample sizes. Various acellular chemical assays which assess the OP by replicating intrinsic biological
mechanisms were therefore developed as alternatives. These assays are generally divided in two categories. The OP
analysis approaches in the 1st category directly probe the generation of ROS during redox cycling reactions in presence
of PM, such as the measurement of $H_2O_2$ and ·OH production in surrogate lung fluid (SLF) (Vidrio et al., 2009;Shen
et al., 2011;Charrier et al., 2014;Ma et al., 2015), and $H_2O_2$ and ·OH production in dithiothreitol (DTT) (Yu et al.,
2018;Xiong et al., 2017;Chung et al., 2006;Kumagai et al., 2002). The assays in 2nd category utilize the consumption
of antioxidants such as AA (Visentin et al., 2016;Weichenthal et al., 2016b) and GSH (Künzli et al., 2006;Szigeti et
al., 2016), or surrogates of cellular reductants such as DTT (Verma et al., 2014;Cho et al., 2005), as the OP indicator.
Analyzing each PM sample for all of these chemical assays is also time-consuming. To address this concern, we have
previously developed an automated OP analysis instrument named SAMERA – Semi-Automated Multi-Endpoint
ROS-activity Analyzer, which can measure five most commonly used OP endpoints (i.e. consumption rate of AA and
GSH in SLF, $OP^{AA}$ and $OP^{GSH}$ respectively; consumption rate of DTT, $OP^{DTT}$, and generation rate of ·OH in SLF and
DTT, $OP^{OH-SLF}$ and $OP^{OH-DTT}$) for a PM extract in less than 3 hours (Yu et al., 2020). These acellular endpoints have
been widely implemented by various researchers for assessing the oxidative properties of $PM_{2.5}$. However, there has
not been a single study which has systematically compared the responses of all of these chemical assays in a single
investigation.
Although OP is proposed as an integrative $PM_{2.5}$ property, purportedly combining the individual and synergistic
actions of its many active components, there have been limited attempts to integrate it in the large-scale
epidemiological studies. This is because, unlike other PM properties such as mass, sulfate, nitrate etc., the OP



measurements in different geographical regions have been relatively sparse. Moreover, before integrating OP in the
epidemiological studies, it is important that we investigate the differences of its spatiotemporal distribution with other
commonly measured PM properties such as mass. An understanding of the temporal variation of OP in a specific
environment could be helpful in time series studies of short-term effects, while the spatial variation of OP can aid in
studying the long-term health effects of $PM_{2.5}$ exposure among different regions (Yang et al., 2015a). Globally, the
spatiotemporal profiles of OP have been characterized for some geographical regions such as Los Angeles Basin
(Saffari et al., 2014, 2013), Denver (Zhang et al., 2008), Atlanta (Fang et al., 2016;Verma et al., 2014) in US, Ontario
(Canada) (Jeong et al., 2020;Weichenthal et al., 2019;Weichenthal et al., 2016a), Netherland (Yang et al., 2015a;Yang
et al., 2015b), and some coastal cities of Bohai [Jinzhou, Tianjin and Yantai (Liu et al., 2018)] and Beijing (Yu et al.,
2019;Liu et al., 2014) in China. Some of these studies have substantially contributed in enhancing our understanding
of the role of OP in the PM-induced health effects (Fang et al., 2016;Tuet et al., 2016;Abrams et al., 2017;Weichenthal
et al., 2016a;Yang et al., 2016;Bates et al., 2015). However, despite including many cities ranked high in terms of the
air pollution [e.g. Indianapolis (Rosenthal et al., 2008), Chicago (Dominici et al., 2003), St. Louis (Sarnat et al., 2015),
Detroit (Zhou et al., 2011), Cincinnati (Kaufman et al., 2019), and Cleveland (Kumar et al., 2013)], the midwestern
region of the United States is an understudied region in terms of assessing the oxidative levels of ambient $PM_{2.5}$.
Here, we investigate the detailed spatiotemporal profiles of ambient $PM_{2.5}$ OP in the midwestern United States.
Simultaneous ambient $PM_{2.5}$ samples were collected from five different sites in the Midwest US. The automated
instrument – SAMERA facilitated the measurement of OP on our large bulk of $PM_{2.5}$ samples (N = 241) collected
from all the sites, which were extracted in both water and methanol separately. This paper mainly discusses the
spatiotemporal distribution of the mass concentration and OP of $PM_{2.5}$ measured by five different endpoints in the
Midwest US. Correlations of OP with PM chemical composition and source apportionment analysis of $PM_{2.5}$ OP will
be presented in our subsequent publications. Our paper presents the results from probably one of the most
comprehensive OP analysis campaigns, combining five different acellular OP endpoints measured on both water- and
organic-soluble extracts.
**2 Experimental methods**
2.1 Sampling campaign
Simultaneous sampling in five different sites spread across three states (i.e. Illinois, Indiana and Missouri) was
conducted every week for this project in the Midwest US. The locations of the sampling sites are shown in Figure 1.
Champaign (CMP) and Bondville (BON) sites are paired sites representing the urban (roadside) and rural environment
of Champaign County, IL, respectively; while three major city sites [i.e. Chicago (CHI), Indianapolis (IND) and St.
Louis (STL)] are representatives of urban background regions of Chicago, Indianapolis and St. Louis, respectively.
CMP is located on a parking garage in the campus of University of Illinois at Urbana-Champaign, and is adjacent to
a 2-lane (both ways) road (i.e. University Avenue). This site is surrounded by the university facilities and is impacted



by traffic emissions from adjacent road. The site is about 1 km from downtown Champaign and is surrounded by
dense housing and business development.
BON is a rural site, 15 km west of downtown Champaign, and is also a part of the IMPROVE (Interagency Monitoring
of Protected Visual Environments) monitoring program. The station is managed by the Illinois State Water Survey,
and is surrounded by intensively managed agricultural fields. The major highways (I-57 and I-74) are at least 6 km
north and east of this site, respectively.
CHI site is located on a dormitory building – Carman hall in Illinois Institute of Technology (IIT) campus, Chicago,
IL. This site is ~500 m away from a two-way 6-lane (including an emergency lane) interstate highway I-90/94, 1.5
km west of Lake Michigan and 5 km south of downtown Chicago. The highway I-90/94 has an annual average daily
traffic flow of 300,000 vehicles per day, and heavy-duty vehicles account for ~10% in the traffic fleet (Xiang et al.,
2019). The site is situated in the mixed commercial and residential area of Chicago, and therefore the emissions from
both traffic mixed with residential and commercial activities are expected.
IND site is located inside the campus of School of Public Health, Indiana University – Purdue University Indianapolis
(IUPUI). This site is close to downtown Indianapolis (2 km southeast of IND site) and a two-way 4-lane interstate
highway I-65 (1 km northeast of IND site). The site is surrounded by miscellaneous facilities of IUPUI and Riley
Hospital, therefore the sources of ambient aerosols at IND site may include vehicular emissions from highway, and
emissions from residential and commercial activities related to miscellaneous university and hospital operations.
STL site is located 3 km north of downtown St. Louis, MO. This site is 230 m west of the interstate I-44/70 and 1.2
km west of Mississipi River. It is also surrounded by several industries for steel processing, zinc smelting and copper
production (Lee et al., 2006). Therefore, a significant portion of metals in PM at this site is supposed to be from
industrial emissions. The urban activities in downtown St. Louis as well as traffic emissions from highway vehicles
and river boating are also potential sources of $PM_{2.5}$ at this site.
The sampling period involved four seasons starting from May 22, 2018 to May 30, 2019. Integrated ambient $PM_{2.5}$
samples were collected simultaneously for three continuous days from all the sites. Each site was instrumented with
a High-volume (Hi-Vol) air sampler equipped with $PM_{2.5}$ inlet (flow rate = 1.13 $m^3$/min; Tisch Environmental; Cleves,
OH). All the samplers were equipped with a timer to enable automatic start of the sampling on each Tuesday 0:00,
and turn-off on each Friday 0:00. After the sampled filters were collected on Friday (before noon), new filters were
loaded in the filter holder to start next run of sampling. We used quartz filters (Pall TissuquartzTM, 8"×10") for
collecting $PM_{2.5}$. The filters were prebaked at 550 °C for 24 hours before sampling. Total 241 filters were collected
during the whole campaign (44 from CHI, 47 from STL, 54 from IND, 51 from CMP and 45 from BON). We also
collected field blank filters (N = 10 from each site) once in every five weeks by placing a blank quartz filter in filter
holder of the sampler for 1 hour but without running the pump.
All filters were weighed before and after sampling using a lab-scale digital balance (0.2 mg readability, Sartorius
A120S, Götingen, Germany) for determining the $PM_{2.5}$ mass loading on each filter. Prior to each weighing, filters
were equilibrated in a constant temperature (24 °C) and relative humidity (50 %) room for 24 hours. After sampling,



the filters were individually wrapped in prebaked (550 °C) aluminum foils and stored in a freezer at -20 °C before
analysis. More information on sampling including the exact dates of sampling are provided in Table S1 in the
supplemental information (SI).
2.2 Sample extraction protocol
Sample extraction protocol for OP analysis was determined by the requirement to keep a relatively constant
concentration of $PM_{2.5}$ in the liquid extracts. This is due to non-linear response of certain OP endpoints with $PM_{2.5}$
mass in the extracts (Charrier et al., 2016). Thus, fraction of the filter and the volume of water used for extraction
were varied depending on the $PM_{2.5}$ mass loading on each Hi-Vol filter. For the analyses of water-soluble OP, a few
(usually 3-5) circular sections (16-25 mm diameter) were punched from the filter and immersed into 15-20 mL of
deionized Milli-Q water (DI, resistivity = 18.2 MΩ/cm). The volume of water was adjusted to achieve ~100 µg of
total $PM_{2.5}$ per mL of DI. The vials containing filter sections suspended in the DI were sonicated in an ultrasonic water
bath for 1 hour (Cole-Palmer, Vernon-Hills, IL, US). These suspensions were then filtered through a 0.45 µm PTFE
syringe filter to remove all water-insoluble components including filter fibers. 10.5 mL of these filtered extracts were
separated and diluted with DI to 15 mL. These diluted extracts were then kept in the sample queue of SAMERA for
OP analyses. SAMERA withdraws different volume of these extracts into the reaction vials (RVs) for each OP
measurement, i.e. 3.5 mL for $OP^{AA}$, $OP^{GSH}$ and $OP^{OH-SLF}$, and 2.1 mL for $OP^{DTT}$ and $OP^{OH-DTT}$ measurements, all of
which were further diluted to 5 mL in the RVs. Thus, the concentrations of $PM_{2.5}$ in RVs for SLF-based (i.e. $OP^{AA}$,
$OP^{GSH}$ and $OP^{OH-SLF}$) and DTT-based (i.e. $OP^{DTT}$ and $OP^{OH-DTT}$) assays were maintained constant at 50 µg/mL and 30
µg/mL (±1%), respectively.
For methanol-soluble OP measurements, another fraction from each filter having the same area as used for the water-
soluble $PM_{2.5}$ extraction was punched and extracted in 10 mL of methanol. After sonication for 1 hour, the suspensions
were filtered through 0.45 µm PTFE syringe filter. The filtered extracts were then concentrated to less than 50 µL
using a nitrogen dryer to evaporate methanol, and were subsequently reconstituted into 15-20 mL of DI, diluted and
analyzed for OP in the same way as water-soluble extracts.
2.3 OP analysis
OP activities of $PM_{2.5}$ extracts were analyzed using SAMERA. The setup and operation protocol of SAMERA has
been discussed in detail in Yu et al. (2020). Briefly, the analysis of all OP endpoints for each extract was conducted
in two stages: SLF-based endpoints were analyzed first, while DTT-based assays were conducted in the second stage.
For measuring $OP^{AA}$ and $OP^{GSH}$, 3.5 mL of the extract was mixed with 0.5 mL SLF and 1 mL of 0.5 M potassium
phosphate buffer (K-PB) in an RV. At certain time intervals (i.e. 5, 24, 43, 62 and 81 minutes), two small aliquots of
the reaction mixture were withdrawn and dispensed into two measurement vials (MV1 and MV2) separately. The
mixture in MV1 was diluted by DI, and was directly injected into a liquid waveguide capillary cell (LWCC-3100;
World Precision Instruments, Inc., Sarasota, FL, USA) coupled to an online spectrophotometer (Ocean Optics, Inc.,
Dunedin, FL, USA), which measured the absorbance at 265 nm (signal from AA) and 600 nm (background) for
determining the concentration of AA. 1.6 mL of o-phthalaldehyde (OPA) was added into the reaction mixture



contained in MV2 to react with GSH, which forms a fluorescent product. The final mixture in MV2 was then pushed
through a flow cell equipped in a Horiba Fluoromax-4 spectrofluorometer (Horiba Scientific, Edison, NJ, USA), and
the fluorescence was measured at excitation/emission wavelength of 310 nm/427 nm. Simultaneously with the
preparation of the reaction mixture for $OP^{AA}$ and $OP^{GSH}$ analyses, 3.5 mL of the extract was mixed with 0.5 mL SLF
and 1 mL of 50 mM K-PB buffered disodium terephthalate (TPT) (pH = 7.4) in another RV2. TPT captures ·OH
generated in the reaction and forms another fluorescent product 2-hydroxyterephthalic acid (2-OHTA). Small aliquots
of this reaction mixture were withdrawn into MV2 at selected time intervals (10, 29, 48, 67 and 86 minutes), diluted
by DI, and injected into the flow cell of the spectrofluorometer for measuring fluorescence at the same wavelengths
as used for GSH measurement (i.e. 310 nm excitation/427 nm emission). The concentration of 2-OHTA was
determined by calibrating various concentrations (10-500 nM) of 2-OHTA standards, and the generation rate of ·OH
was determined as the formation rate of 2-OHTA divided by a yield factor (0.35) (Son et al., 2015).
Both RVs and MVs were flushed with DI after all SLF-based endpoints were analyzed, and DTT-based assays started
immediately after this cleaning. Similar to the first step of SLF assay, 2.1 mL of the diluted $PM_{2.5}$ extract was mixed
with 1 mL of 50 mM TPT, 1.4 mL of DI and 0.5 mL of 1 mM DTT in an RV. At certain time intervals (i.e. 5 min, 17
min, 29 min, 41 min and 53 min), two small aliquots of this reaction mixture were withdrawn and diluted with DI in
MV1 and MV2 separately for the measurement of DTT and ·OH, respectively. DTNB was added into MV1 to capture
residual DTT. The final mixture in MV1 was pushed through LWCC to measure the absorbance at 412 nm, while the
mixture in MV2 was pushed through flow cell of the spectrofluorometer for fluorescence measurement (310 nm
excitation/427 nm emission), respectively. The system was again cleaned by flushing DI to RVs, MVs, LWCC and
flow cell of the spectrofluorometer for the next run. Once in a week, we conducted thorough cleaning of the entire
system, by replacing all chemicals and samples first with methanol followed by DI, and running the program script
10 times with each solvent.
2.4 Quality Control/Quality Assurance
One field blank filter extract along with a DI blank were used as the negative controls for each set of $PM_{2.5}$ samples
analyzed in a batch (usually ~10). Selected metals and organic compounds that are known to be sensitive for different
OP endpoints, i.e. Cu(II) for $OP^{AA}$ and $OP^{GSH}$, Fe(II) for $OP^{OH\text{-}SLF}$, phenanthraquinone for $OP^{DTT}$ and 5-hydroxy-1,4-
naphthoquinone for $OP^{OH\text{-}DTT}$, were used as the positive control, and were analyzed weekly with $PM_{2.5}$ samples to
ensure the stability of SAMERA and correct for any possible drift.
The average and standard deviation of OP of negative and positive controls are shown in Table 1. Our previous study
on the development of SAMERA (Yu et al., 2020) reported the values of OP for negative controls, as $0.17 \pm 0.07$
μM/min for $OP^{AA}$, $0.37 \pm 0.06$ μM/min for $OP^{GSH}$, $4.57 \pm 1.21$ nM/min for $OP^{OH\text{-}SLF}$, $0.65 \pm 0.02$ μM/min for $OP^{DTT}$
and $-0.38 \pm 0.24$ μM/min for $OP^{OH\text{-}DTT}$. Consistency of our current results for negative controls with those reported
earlier, and a low coefficient of variation (CoV) obtained for the positive controls (1.1 – 11.8%), ensured a good
quality assurance for the overall OP analysis. We blank corrected all OP values of ambient samples by subtracting the





averaged field blank measurements. After blank correction, the OP values below detection limit were replaced with
half of the detection limits for the corresponding OP endpoint.
2.5 Statistical analysis
To assess spatiotemporal variability in both OP and $PM_{2.5}$ mass, we compared their differences among all sites and
seasons using one-way analysis of variance (ANOVA) test, and different pairs (i.e. pairs of different sites or seasons)
were compared by Fisher's least significant difference (LSD) post-hoc test. The significant and highly significant
differences were considered by one-way ANOVA when $P < 0.05$ and $P < 0.01$, respectively. Pearson's correlation
coefficient (r) for single linear regression was computed to determine the correlation of OP between different sites,
between water-soluble and methanol-soluble OP, between OP and $PM_{2.5}$, as well as the intercorrelation among
different endpoints for each site. Since several OP endpoints (e.g. $OP^{AA}$, $OP^{GSH}$ and $OP^{DTT}$) were abnormally elevated
in the week of July $4^{th}$ (Independence Day celebration; discussed in section 3.2), we removed this week's sample from
our regression analysis to avoid any bias caused by this episodic event. Site-to-site comparisons were performed by
calculating the coefficient of divergence (COD) of mass concentration and volume-normalized OP (i.e. OPv) for all
site pairs, as follows:
$$CoD = \sqrt{\frac{1}{N}\sum_{i=1}^{N}\left(\frac{c_{ij} - c_{ik}}{c_{ij} + c_{ik}}\right)^2}$$

where: $c_{ij}$ and $c_{ik}$ are the $PM_{2.5}$ mass or OPv measured in the same week $i$ at sites $j$ and $k$, respectively; N is the number
of the comparable sample pairs for sites $j$ and $k$. COD ranges from 0 to 1. A larger COD (closer to 1) indicates more
spatial heterogeneity between the sites, while a smaller COD (closer to 0) implies spatial homogeneity. One-way
ANOVA test was conducted in Matlab R2019a, while other statistical analyses were carried out using Excel.
**3 Results and Discussion**
3.1 $PM_{2.5}$ mass concentration
Figure 2 shows the time series of three-days averaged $PM_{2.5}$ mass concentration at five sampling sites, while the
seasonal averages are shown in Table 2. The mass concentrations ranged from 2.0 to 21.7 μg/m³ across all sites, and
the median was 11.0 μg/m³. These results are comparable with previous studies on $PM_{2.5}$ in Midwest US cities, e.g.
St. Louis (3.9 - 48.6 μg/m³) (Lee et al., 2006), Chicago (median 9.4 – 10.7 μg/m³) (Milando et al., 2016), Detroit (0.6
– 56.2 μg/m³, median 14.4 – 17.6 μg/m³) (Gildemeister et al., 2007), Bondville (2.1 – 36.5 μg/m³, median 9.5 μg/m³)
and selected cities in Iowa (e.g. Cedar Rapids, Des Moines and Davenport) (8.4 – 11.6 μg/m³) (Kundu and Stone,
2014). Generally, the more urbanized sites of our study (i.e. CHI, STL and IND) showed slightly higher mass
concentrations (5.7 – 21.7 μg/m³) compared to the smaller cities like CMP and its rural component (i.e. BON) (2.0 –
20.2 μg/m³). The highest mass concentrations were recorded at CHI (during winter) and STL (during summer), while
BON exhibited the lowest concentrations in all seasons, except fall when the mass concentrations were lowest at CMP.





Other than these minor variations, the PM$_{2.5}$ mass concentrations are both spatially and temporally homogeneous in
the Midwest US with no significant seasonal differences.
3.2 Time series of PM$_{2.5}$ OP
Time series of both mass- and volume-normalized OP (OPm and OPv, respectively) at all the sites are shown in Figure
3 (water-soluble OP) and Figure 4 (methanol-soluble OP). Generally, OP for both water- and methanol-soluble
extracts showed much more spatiotemporal variability than the PM$_{2.5}$ mass in the Midwest US. For water-soluble OP,
we observed significant spatial variability for SLF-based endpoints (i.e. OP$^{AA}$, OP$^{GSH}$ and OP$^{OH-SLF}$) in both mass- and
volume-normalized results (Figure 3a-c). CMP showed a substantially higher water-soluble OP than other sites for
these endpoints. In the temporal trend, SLF-based endpoints showed higher levels during summer compared to other
seasons at most sites. A significant temporal variation was observed for CMP with several spikes in the OP activities
throughout the year, most prominently for OP$^{AA}$. The peak in the week of July 3 were observed for multiple endpoints
(e.g. OP$^{AA}$, OP$^{GSH}$ and OP$^{DTT}$) at most sites, which is attributed to the emissions from firecrackers on Independence
Day (July 4) celebrations (Yu et al., 2020;Puthussery et al., 2018). In comparison to SLF-based endpoints, mass- and
volume-normalized DTT-based OP (i.e. OP$^{DTT}$ and OP$^{OH-DTT}$) showed lesser spatial variations (Figure 3d-e). The
spatiotemporal variations for the methanol-soluble OP endpoints (e.g. OP$^{AA}$, OP$^{GSH}$, OP$^{DTT}$ and OP$^{OH-DTT}$) seem to be
lesser than the corresponding water-soluble OP (Figure 4a-b, d-e). However, methanol-soluble OP$^{OH-SLF}$ showed a
significant seasonal variability with substantially higher levels in summer at most sites, and a marginal spatial
variability with slightly higher activities at CHI during summer (Figure 4c). The spatiotemporal trends for mass- and
volume-normalized OP activities were very similar for both water and methanol extracts.
A comparison of the ranges of OP endpoints observed in our study and previous investigations has been briefly
provided in SI (Table S2). For water-soluble PM$_{2.5}$ in our study, OP$^{AA}$m ranged from 0.002 to 0.077 nmol·min$^{-1}$·µg$^{-1}$,
which is within the ranges reported from previous studies conducted in Europe (Künzli et al., 2006;Szigeti et al.,
2016;Godri et al., 2011) and India (Mudway et al., 2005). However, our range of OP$^{AA}$v (0.012 – 0.908 nmol·min$^{-1}$·m$^{-3}$) is much lower than that reported by Fang et al. (2016) (0.2 – 5.2 nmol·min$^{-1}$·m$^{-3}$), probably because of a different
protocol used in their study, which involved only AA in the assay. The median of water-soluble OP$^{GSH}$m (0.007
nmol·min$^{-1}$·µg$^{-1}$) is also comparable with the average of those reported (0.0041 – 0.0083 nmol·min$^{-1}$·µg$^{-1}$) in previous
studies (Mudway et al., 2005;Künzli et al., 2006;Godri et al., 2011). Similarly, the median of OP$^{OH-SLF}$m (0.142
pmol·min$^{-1}$·µg$^{-1}$) is comparable to the averages reported by Vidrio et al. (2009) (0.253 pmol·min$^{-1}$·µg$^{-1}$) and Ma et al.
(2015) (0.092 pmol·min$^{-1}$·µg$^{-1}$). The median of OP$^{DTT}$m (0.014 nmol·min$^{-1}$·µg$^{-1}$) of our samples is significantly lower
than the medians or averages reported from most studies conducted in US (0.019 – 0.041 nmol·min$^{-1}$·µg$^{-1}$) (Cho et al.,
2005;Charrier and Anastasio, 2012;Gao et al., 2020;Hu et al., 2008;Fang et al., 2015). Similarly, the median of our
OP$^{DTT}$v (0.150 nmol·min$^{-1}$·m$^{-3}$) is lower compared to several studies in Southeast US (0.19 – 0.31 nmol·min$^{-1}$·m$^{-3}$)
(Gao et al., 2017;Gao et al., 2020;Fang et al., 2015), but closer to one study conducted in Southwest US (0.14
nmol·min$^{-1}$·m$^{-3}$) (Hu et al., 2008). The range of water-soluble OP$^{OH-DTT}$v of our samples is quite large (0.004 – 3.565
pmol·min$^{-1}$·m$^{-3}$); however, there is no previous data to compare it, other than reported in the studies conducted by our
own group (Xiong et al., 2017;Yu et al., 2018), which were based on a much smaller sample size (N = 10) and limited





spatial extent (single site) and thus resulting into a much narrower range ($0.2 - 1.1$ pmol·min$^{-1}$·m$^{-3}$). Compared to
water, only a handful of studies on PM OP$^{DTT}$ used methanol as the PM extraction solvent, while no previous literatures
have investigated the OP of PM for other endpoints. The medians of our methanol-soluble OP$^{DTT}$m ($0.021$ nmol·min$^{-1}$·µg$^{-1}$)
and OP$^{DTT}$v ($0.234$ nmol·min$^{-1}$·m$^{-3}$) are slightly lower than the medians or averages reported in previous studies
in the Southeast US ($0.027 - 0.034$ nmol·min$^{-1}$·µg$^{-1}$ and $0.28 - 0.30$ nmol·min$^{-1}$·m$^{-3}$, respectively for OP$^{DTT}$m and
OP$^{DTT}$v) (Verma et al., 2012;Gao et al., 2017;Gao et al., 2020), which is consistent with the trend for water-soluble
OP$^{DTT}$ (i.e. lower levels of our samples than reported previously at other sites).
3.3 Spatiotemporal variation in PM$_{2.5}$ OP
*Water-soluble PM$_{2.5}$ OP*
Seasonally averaged OPm and OPv of water-soluble PM$_{2.5}$ at different sites are shown in Figure 5. Differences in both
OPm and OPv among different seasons or sites were determined by one-way ANOVA and the results are listed in SI,
Table S3. Seasonally, highest OP activities were generally observed in summer, while the lowest activities usually
occurred in winter. For example, OP$^{AA}$v and OP$^{GSH}$v activities had highest levels in summer and lowest levels in
winter at CMP and BON, as verified by 1-way ANOVA ($P < 0.05$). Similarly, significantly higher OP activities ($P <$
$0.01$ for most cases) were observed for both OP$^{OH-SLF}$m and OP$^{OH-SLF}$v at all five sites in summer, while winter showed
significantly lower levels ($P < 0.05$). For DTT-based endpoints, OP$^{OH-DTT}$m and OP$^{OH-DTT}$v also showed higher values
in summer at CHI, IND and CMP ($P < 0.01$). However, OP$^{DTT}$ exhibited limited temporal variation at most sites with
only slightly higher OP$^{DTT}$m and OP$^{DTT}$v observed in summer at BON ($P < 0.05$). The seasonal trend of mass- and
volume-normalized activities were nearly identical for all endpoints, indicating a marginal effect of PM$_{2.5}$ mass
concentration in the temporal variation of OP.
The temporal variation trend of OP$^{DTT}$ in this study does not correspond with previous studies conducted in Southwest
and Southeast US. For the Southeast US, Verma et al. (2014) found significantly higher OP$^{DTT}$v in winter (December,
2012) compared to summer (June to August, 2012), and this difference was even more pronounced in mass-normalized
OP. Saffari et al. (2014) also observed higher OP$^{DTT}$ activities of quasi-ultrafine particles (PM$_{0.25}$) in fall and winter
seasons for the Southwest US (Los Angeles Basin), and attributed this trend to the partitioning of redox-active semi-
volatile organic compounds to particle phase in colder seasons. However, the trend of OP$^{AA}$ in our study is in
agreement with another study in Southeast US using OP$^{AA}$ as the endpoint (Fang et al., 2016), which showed higher
OP$^{AA}$ in warmer seasons (i.e. summer and fall) than winter. There is no previous literature available on the
spatiotemporal trends of other OP endpoints in US, to which we can compare our results.
Spatially, there seems higher variability in the SLF-based endpoints, i.e. OP$^{AA}$ and OP$^{GSH}$ than the DTT-based
endpoints (OP$^{DTT}$ and OP$^{OH-DTT}$). Highest OP$^{AA}$ and OP$^{GSH}$ activities (both mass- and volume-normalized) occurred
at the roadside site CMP (as confirmed by 1-way ANOVA test; $P < 0.01$) in most seasons (except winter for OP$^{AA}$v),
while STL and IND had the lowest OP$^{AA}$ and OP$^{GSH}$. OP$^{OH-SLF}$ was more spatially uniformly distributed than OP$^{AA}$
and OP$^{GSH}$; significantly higher OP$^{OH-SLF}$m and OP$^{OH-SLF}$v were observed at CMP only in summer and spring ($P <$
$0.05$). For the DTT-based endpoints, OP$^{DTT}$v was only marginally higher at CHI in winter, and at CMP in summer



and spring. Other than that, no significant differences were observed for $OP^{DTT}v$ among various sites. The spatially
uniform pattern for $OP^{DTT}v$ is consistent with Verma et al. (2014) which found limited spatial variation for $OP^{DTT}v$ in
the Southeast US. In contrast, there was significant variation in the $OP^{DTT}m$ with elevated levels at CMP ($P < 0.01$) in
all seasons. Interestingly, the $OP^{OH-DTT}$ endpoint showed more spatial variability and was generally lowest at CMP ($P
< 0.05$) – the site which showed highest levels for all other OP endpoints. It implies that although $OP^{DTT}$ and $OP^{OH-
DTT}$ endpoints are measured in the same DTT assay, different chemical components play differential roles in these
endpoints. We found very similar spatial patterns of mass- and volume-normalized OP activities for most endpoints,
again indicating only a marginal role of $PM_{2.5}$ mass concentrations in causing the spatial variability in OP levels.
*Methanol-soluble $PM_{2.5}$ OP*
Seasonal averages of methanol-soluble $PM_{2.5}$ OPm and OPv are shown in Figure 6.  Compared to water-soluble OP,
most OP endpoints in the methanol-soluble extracts showed weaker seasonal variations, as also indicated by relatively
lower F-values [median of $F = 1.61$ (Table S4a), compared to 2.71 for the water-soluble OP endpoints (Table S3a)].
Similar to water-soluble OP, highest activities for the methanol-soluble OP were generally observed in summer. For
example, highest values of $OP^{AA}$ and $OP^{DTT}$ were observed in summer at CMP and BON ($P < 0.05$) for both mass-
and volume-normalized activities. $OP^{OH-SLF}m$ and $OP^{OH-SLF}v$ peaked in summer at BON ($P < 0.01$), but in fall at IND
($P < 0.05$). $OP^{OH-DTT}m$ and $OP^{OH-DTT}v$ were also elevated in summer at CHI ($P < 0.01$), but showed marginal seasonal
variations at other sites. In contrast, $OP^{GSH}$ showed a rather homogeneous seasonal distribution at all sites, except
slight elevation of $OP^{GSH}m$ in fall at STL and IND ($P < 0.05$).
The spatial variations in OP were also weaker for the methanol-soluble extracts in comparison to water-soluble
extracts [median of $F = 1.96$ (Table S4b), compared to 4.52 for the water-soluble OP endpoints (Table S3b)]; however,
some spikes were observed at certain sites in different seasons. Substantially higher $OP^{AA}v$ occurred at CHI ($P < 0.05$)
in winter and spring, while no significant differences were observed for $OP^{AA}m$ among different sites in any other
season. $OP^{GSH}v$ was elevated at CHI and CMP during winter and spring ($P < 0.05$), while CMP showed elevated
$OP^{GSH}m$ in all seasons ($P < 0.05$). In summer and winter, $OP^{OH-SLF}$ peaked at CHI ($P < 0.05$) for both mass- and
volume-normalized levels. $OP^{OH-DTT}m$ and $OP^{OH-DTT}v$ also peaked at CHI ($P < 0.05$) in summer. The lowest levels of
$OP^{OH-DTT}$ were again found at CMP in all seasons, which is consistent with the trend for water-soluble $OP^{OH-DTT}$. In
contrast, $OP^{DTT}$ showed spatially homogeneous distribution across all seasons, with marginally elevated values of
$OP^{DTT}v$ at STL during fall and winter ($P < 0.05$). The spatiotemporal trends were again very similar between mass-
and volume-normalized methanol-soluble OP activities except few cases discussed here.
3.4 Comparison of water-soluble and methanol-soluble OP
To assess the effect of solvent on the OP response, we computed the ratio of methanol-soluble OPv to water-soluble
OPv ($M/W^{OP}$) for all samples, and plotted it for the individual sites in Figure 7. As shown in the figure, methanol-
soluble extracts generally showed greater response for most of the OP endpoints than the water-soluble extracts, with
medians of $M/W^{OP}$ being either close or greater than 1. The medians for $M/W^{OP}$ for $OP^{GSH}v$ and $OP^{DTT}v$ were closer
to 1 at many sites ($0.6 – 1.3$ for $OP^{GSH}v$; and $1.1 – 1.9$ for $OP^{DTT}v$), while significantly greater than 1 for the other



three endpoints (OP$^{AA}$v, OP$^{OH-SLF}$v and OP$^{OH-DTT}$v). The only exception to this trend was for OP$^{AA}$v at CMP, where
significantly lower levels of methanol-soluble OP than water-soluble OP were observed (median of M/W$^{OP}$ = 0.7 for
OP$^{AA}$v at CMP). Our previous studies analyzing the chemical composition of PM collected at CMP have shown an
elevated level of Cu (up to 60 ng/m$^3$) at this site (Wang et al., 2018;Puthussery et al., 2018), compared to the typical
range (4 – 20 ng/m$^3$) at most urban sites in US (Buzcu-Guven et al., 2007;Kundu and Stone, 2014;Lee and Hopke,
2006;Hammond et al., 2008;Baumann et al., 2008;Milando et al., 2016). Although water-soluble Cu has been shown
as the most important contributor to OP$^{AA}$ (Fang et al., 2016;Ayres et al., 2008;Visentin et al., 2016), Lin and Yu
(2020) reported a strong antagonistic interaction of Cu with imidazole and pyridine, both of which are alkaloid
compounds (i.e. reduced organic nitrogen compounds), for oxidizing AA. The unprotonated nitrogen atom in alkaloids
tends to chelate Cu, thus reducing its reactivity with AA. Since many of the alkaloid compounds are water-insoluble
but methanol-soluble, it is possible that these compounds are efficiently extracted in methanol, causing the apparently
lower levels of methanol-soluble OP$^{AA}$ compared to the water-soluble OP$^{AA}$ at CMP.
The medians of M/W$^{OP}$ were very high for both ·OH based endpoints (i.e. OP$^{OH-SLF}$ and OP$^{OH-DTT}$v) (2.1 – 3.8 for
OP$^{OH-SLF}$v and 1.4 – 1.9 for OP$^{OH-DTT}$v), indicating that methanol is able to more efficiently extract the redox-active
components driving the response of these OP endpoints. We suspect that one of such components could be organic-
complexed Fe. As a Fenton reagent, Fe can catalyze the transfer of electrons from H$_2$O$_2$ to ·OH (Held et al., 1996).
The generation of ·OH is further enhanced by the complexation of Fe with organic species (Wei et al., 2018;Gonzalez
et al., 2017;Xiong et al., 2017;Yu et al., 2018). In a previous study conducted at our CMP site, Wei et al. (2018) found
a significant fraction of Fe complexed with hydrophobic organic species (28 ± 22 %). That study also reported a
substantially higher ratio of Fe concentration in 50 % methanol to that in water (1.42 ± 0.19), which showed some
seasonality (1.97 ± 0.17 during winter and 1.33 ± 0.20 in summer). This seasonal pattern of Fe solubility in methanol
versus water is consistent with the time series of M/W$^{OP}$ for OP$^{OH-SLF}$v at most sites (showing higher values in winter
than summer; SI Table S5), which further corroborated that Fe complexed with hydrophobic organic fraction of PM$_{2.5}$
could be majorly responsible for the OP$^{OH-SLF}$v and OP$^{OH-DTT}$v in the methanol extracts. However, detailed chemical
characterization will be needed to confirm these hypotheses, which will be explored in our subsequent publications.
We also calculated Pearson's r for the regression between respective water-soluble and methanol-soluble OP endpoints
for individual sites, which are shown in Table 3. OP$^{DTT}$v showed some good correlation between two extraction
protocols (r = 0.43 – 0.74 except at STL), while correlations were generally poor (r < 0.60) for other four endpoints
(i.e. OP$^{AA}$v, OP$^{GSH}$v, OP$^{OH-SLF}$v and OP$^{OH-DTT}$v). It indicates that the components driving the response of OP$^{DTT}$ could
be more uniformly extracted in both water and methanol. However, there are additional water-insoluble species driving
the response of OP$^{AA}$v, OP$^{GSH}$v, OP$^{OH-SLF}$v and OP$^{OH-DTT}$v, which are more efficiently extracted in methanol than
water.



3.5 Site-to-site comparison of OP and mass concentration of PM$_{2.5}$
To further evaluate the spatial trend of OP across the Midwest US region, we calculated both COD and correlation
coefficients (Pearson's r) for different site pairs, which are shown in Figure 8 (mass concentrations and water-soluble
OP of PM$_{2.5}$), and Figure 9 (methanol-soluble PM$_{2.5}$ OP).
*PM$_{2.5}$ mass concentration and water-soluble PM$_{2.5}$ OP*
PM$_{2.5}$ mass concentrations showed low levels of COD (0.13 – 0.25, median: 0.20), confirming a spatially
homogeneous distribution of PM$_{2.5}$ as indicated earlier (Figure 8a). Conversely, we observed generally higher CODs
for all water-soluble OPv endpoints, i.e. OP$^{AA}$v (0.38 – 0.56, median: 0.43), OP$^{GSH}$v (0.28 – 0.51, median: 0.35),
OP$^{OH-SLF}$v (0.30 – 0.40, median: 0.35), OP$^{DTT}$v (0.19 – 0.34, median: 0.25), and OP$^{OH-DTT}$v (0.21 – 0.38, median: 0.27)
(Figure 8b-f). Our results showing a stronger spatial variability in OP than PM mass are largely in agreement with a
recent study (Daellenbach et al., 2020) analyzing a comprehensive dataset for OP in Europe, which showed that both
OPv (measured by DTT, 2′,7′-Dichlorofluorescin Diacetate and AA assays) and PM$_{10}$ mass concentrations were
elevated in the urban environments (e.g. Paris and the Po valley), but PM$_{10}$ was more regionally distributed than OPv.
Interestingly, we found poor correlations for PM$_{2.5}$ among all site pairs (r < 0.60), except IND and BON (r = 0.63). It
implies that despite a homogeneous spatial distribution, emission sources of the chemical species composing PM$_{2.5}$
are different at different sites. The correlations were also weak (r < 0.60 for most cases) for the OP endpoints showing
high CODs, i.e. OP$^{AA}$, OP$^{GSH}$, OP$^{OH-SLF}$ and OP$^{OH-DTT}$, which indicates a more pronounced effect of local point sources
on these OP endpoints compared to the regional sources. In contrast, OP$^{DTT}$v showed stronger correlation (r = 0.48 –
0.76, median: 0.62) for most site pairs. Higher correlations for the DTT activity combined with lower CODs suggests
that the regional sources such as long-range transport or atmospheric processing could have a larger influence on
OP$^{DTT}$ than the local sources.
*Methanol-soluble PM$_{2.5}$ OP*
In comparison to water-soluble PM$_{2.5}$ OP, CODs for the methanol-soluble OP were generally lower (median: 0.21 –
0.35; Figure 9), indicating higher spatial homogeneity of methanol-soluble PM chemical components that are sensitive
to OP. Similar to water-soluble OP$^{DTT}$v, the methanol-soluble OP$^{DTT}$v showed the lowest COD (0.14 – 0.26, median:
0.21) among five endpoints (Figure 9d), which was consistent with Gao et al. (2017) showing a rather low COD (less
than 0.23) for both water-soluble and methanol-soluble OP$^{DTT}$ in Southeast US. Overall, higher correlation coefficients
were observed for the methanol-soluble OP (median: 0.41 – 0.67 for different endpoints) than the corresponding water-
soluble endpoints (median: 0.13 – 0.62). The correlation coefficients were more elevated for certain endpoints such
as OP$^{AA}$v (r = 0.38 – 0.62, median: 0.46) and OP$^{GSH}$v (r = 0.23 – 0.65, median: 0.41) than others. It is possible that
methanol is able to extract more redox-active PM components coming from common emission sources present at these
sites, and thus yielding to an overall lower spatiotemporal variability and better correlation among different sites.



3.6 Correlations of OP with PM$_{2.5}$ mass concentration
Pearson's r and the slope for simple linear regression of volume-normalized OP activities versus PM$_{2.5}$ mass
concentrations were computed for each individual site, and are listed in Table 4. For both water-soluble and methanol-
soluble OP, the endpoints of OP$^{AA}$v, OP$^{OH-SLF}$v and OP$^{OH-DTT}$v were poorly correlated with PM$_{2.5}$ mass (r < 0.60 in
most cases), while OP$^{GSH}$v and OP$^{DTT}$v were moderately-to-strongly correlated with PM$_{2.5}$ mass (r = 0.38 – 0.73 for
OP$^{GSH}$v, and 0.54 – 0.82 for OP$^{DTT}$v, except at STL). The lower correlation of OP$^{AA}$ and higher correlation of OP$^{DTT}$
are consistent with multiple previous studies comparing these endpoints (Visentin et al., 2016;Yang et al.,
2014;Janssen et al., 2014). Decent correlations for OP$^{GSH}$v and OP$^{DTT}$v showed that PM mass concentrations can drive
these endpoints to some extent at few locations. However, it is important to note that despite these good correlations,
the slope of regression for OP vs. PM$_{2.5}$ mass varied a lot among five sampling sites (range for OP$^{GSH}$v is 0.003 –
0.016 nmol/min/μg, and 0.005 – 0.028 nmol/min/μg for OP$^{DTT}$v), indicating substantial spatiotemporal heterogeneity
in the intrinsic potency of the particles to generate ROS at these sites. This is further corroborated by the spatiotemporal
variability of OP$^{GSH}$m and OP$^{DTT}$m at different sites as shown in Figure 5 and 6. Thus, PM$_{2.5}$ mass concentrations have
only a limited role in determining the oxidative levels of the PM$_{2.5}$ at these sites, and OP seems to be largely driven
by the PM chemical composition.
3.7 Intercorrelation among different OP endpoints
We also calculated the correlation coefficient (Pearson's r) for all pairs of different OPv endpoints at each site, which
are listed in Table 5. A high correlation coefficient indicates a common source (or a common pool of chemical
components) driving the response of those OP endpoints. For water-soluble OP, the intercorrelations among different
endpoints were generally poor at urban sites, i.e. CHI, STL, and IND (r < 0.60). Correlations were also poor for nearly
all pairs of methanol-soluble OP at STL and IND, but CHI showed significantly elevated r values among different OP
endpoints (r = 0.59 – 0.82). Compared to more urbanized sites, the correlations were generally higher at the local sites,
i.e. CMP and BON, with r > 0.60 for many pairs of both water-soluble and methanol-soluble OPv. Since both of these
sites are located in smaller cities, the sources of redox-active components probably have lesser complexity compared
to the major city sites, which have multiple and more complex emission sources. For example, CMP is adjacent to a
major road, and thus largely impacted by the vehicular emissions. Similarly, BON being a rural site is largely impacted
by the agricultural emissions with marginal impact from vehicular emissions and other sources such as long-range
transport from surrounding cities (Kim et al., 2005;Buzcu-Guven et al., 2007). Thus, a lack of other major sources
contributing to components, which can drive these endpoints in different directions through their interactions (i.e.
synergistic or antagonistic), leads to the similarity of their responses and hence a good correlation among them at these
two sites. Among all OP endpoints, OP$^{OH-DTT}$v showed poorest correlations with other endpoints except OP$^{OH-SLF}$v,
with which it was correlated at most sites (i.e. CHI, IND, CMP and BON) for the methanol-soluble extracts (r = 0.66
– 0.84). Since both of these endpoints measure the rate of generation of ·OH, it probably indicates a synergistic role
of metals with organic compounds [e.g. Fe with humic-like substances (HULIS), as shown in many previous studies
(Yu et al., 2018;Charrier and Anastasio, 2015;Gonzalez et al., 2017;Wei et al., 2018;Ma et al., 2015)] in partly driving





the response of both of these endpoints. Note, $OP^{OH-DTT}$ is a relatively newly developed assay, and there is hardly any
previous literature on its comparison with other OP endpoints.
Overall, a poor-to-moderate and inconstant intercorrelation trend among different endpoints of both water-soluble and
methanol-soluble OP at most sites indicates that measuring a single endpoint is not enough to represent the overall OP
activity. The diverse range of OP endpoints used in our study could better capture the role of different PM components
and their interactions via different pathways for driving the oxidative levels of the PM in a region.

## 4 Conclusion

We analyzed both water-soluble and methanol-soluble OP of ambient $PM_{2.5}$ in the Midwest US using five different
acellular endpoints, including $OP^{AA}$, $OP^{GSH}$, $OP^{OH-SLF}$, $OP^{DTT}$ and $OP^{OH-DTT}$. The spatiotemporal profiles of all OP
endpoints and $PM_{2.5}$ mass concentration were investigated for one-year timescale from May 2018 to May 2019 using
the Hi-Vol filter samples collected from five Midwest US sites located in urban, rural, and roadside environments.
Compared to homogeneously distributed $PM_{2.5}$ mass, all OP endpoints showed significant spatiotemporal variations
among different seasons and sites. Seasonally, most OP endpoints generally peaked in summer for both water-soluble
and methanol-soluble OP. Spatially, the roadside site showed the highest OP levels for most OP endpoints in water-
soluble extracts, while there were occasional peaks in methanol-soluble extracts at other urban sites. Our results
showed very limited differences in the spatiotemporal profiles between OPm and OPv for most endpoints, indicating
a marginal role of $PM_{2.5}$ mass in causing the spatiotemporal variability of OP.
Comparing the OP for water- and methanol-soluble extracts, we observed significantly higher OP levels in methanol
extracts than the corresponding water-soluble OP activities. This trend was much stronger for ·OH generation
endpoints (i.e. $OP^{OH-SLF}$ and $OP^{OH-DTT}$), indicating a substantial contribution of Fe and its organic complexes, which
could be more efficiently extracted in methanol. In comparison to water-soluble OP, methanol-soluble OP showed
lower spatial heterogeneity, and higher intercorrelations among different endpoints, which is probably attributed to a
more efficient extraction of water-insoluble redox-active species in methanol originated from various emission sources
at different sites.
The correlations of OP with $PM_{2.5}$ mass showed a diverse range, with certain endpoints such as $OP^{AA}$, $OP^{OH-SLF}$ and
$OP^{OH-DTT}$ showing a poor correlation, while other endpoints (i.e. $OP^{GSH}$ and $OP^{DTT}$) showing a moderate-to-strong
correlation. Despite these occasional strong correlations, the sensitivity of all OP endpoints towards mass, indicated
by the slope of OP vs. $PM_{2.5}$ mass as well as the intrinsic OP (OPm), varied substantially for all OP endpoints across
different sites and seasons, showing only a marginal effect of mass concentrations in controlling the oxidative levels
of $PM_{2.5}$. Moreover, relatively poor and inconsistent correlations among different OP endpoints reflected different
pathways of various ROS-active $PM_{2.5}$ components for exerting oxidative stress.
Collectively, the results obtained through our study provides a strong rationale to recommend that the different
endpoints of OP provide useful and additional information than the mass concentrations, which could be relevant to
assess the public health impacts associated with ambient $PM_{2.5}$. Our future studies will explore the contribution of



different chemical components and their emission sources in determining the oxidative levels of ambient $PM_{2.5}$ in the
Midwest US.
*Data availability.* The data on OP and mass concentration of ambient $PM_{2.5}$ samples collected in the Midwest US are
available upon request from the corresponding author.
*Author contribution.* HY: collection of $PM_{2.5}$ samples, measurement of OP, data analysis, manuscript organization
and writing; JVP: collection of $PM_{2.5}$ samples, manuscript editing and revision; YW: collection of $PM_{2.5}$ samples,
manuscript editing and revision; VV: conceptualization of study design and methodology, manuscript organization
and editing, and overall project supervision.
*Competing Interests.* The authors declare that they do not have any competing interests.
*Acknowledgements.* This material is based upon work supported by the National Science Foundation under Grant No.
CBET-1847237. We acknowledge the support from Brent Stephens, Yi Wang, and Will Wetherell for providing us
the access to the site in Chicago, Indianapolis and St. Louis, respectively.

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



**Figures and Tables**
**Table 1**. Average and standard deviation of OP from various control groups (N = 10) analyzed by SAMERA.

| Endpoint | Unit | Negative control | | Positive control | | | |
|---|---|---|---|---|---|---|---|
| | | Average | Standard deviation | Chemical used as positive control | Average | Standard deviation | Coefficient of variation (CoV, %) |
| $OP^{AA}$ | μM/min | 0.18 | 0.07 | 1 μM Cu | 0.34 | 0.04 | 11.8 |
| $OP^{GSH}$ | μM/min | 0.26 | 0.06 | 1 μM Cu | 0.77 | 0.02 | 2.6 |
| $OP^{OH-SLF}$ | nM/min | 7.69 | 1.37 | 2 μM Fe | 13.80 | 0.70 | 5.1 |
| $OP^{DTT}$ | μM/min | 0.48 | 0.07 | 0.2 μM PQ | 1.84 | 0.02 | 1.1 |
| $OP^{OH-DTT}$ | nM/min | 0.55 | 0.07 | 0.2 μM 5-H-1,4-NQ | 15.45 | 1.19 | 7.7 |

**Table 2.** Seasonal averages (± standard deviation) of $PM_{2.5}$ mass concentrations (unit: $μg/m^3$) at our sampling sites.

| | CHI | STL | IND | CMP | BON |
|---|---|---|---|---|---|
| Summer 2018 | 11.2 ± 3.2 | 14.7 ± 3.4 | 11.9 ± 3.5 | 11.4 ± 3.9 | 10.4 ± 2.0 |
| Fall 2018 | 10.9 ± 3.4 | 13.1 ± 3.7 | 11.5 ± 4.2 | 7.5 ± 4.3 | 9.7 ± 3.5 |
| Winter 2018 | 14.6 ± 3.6 | 11.8 ± 2.8 | 11.0 ± 2.7 | 10.0 ± 3.0 | 8.6 ± 3.0 |
| Spring 2019 | 12.6 ± 4.2 | 13.8 ± 4.0 | 12.2 ± 2.1 | 11.6 ± 3.1 | 9.2 ± 2.3 |

**Table 3.** Pearson's correlation coefficient (r) and the associated levels of significance (P) between water-soluble and methanol-soluble OPv for different endpoints at five sampling sites. Correlations with r > 0.60 are shown in **bold**.

| Site | Pearson's r/significance level (P) for OP endpoints | | | | |
|---|---|---|---|---|---|
| | $OP^{AA}$ | $OP^{GSH}$ | $OP^{OH-SLF}$ | $OP^{DTT}$ | $OP^{OH-DTT}$ |
| CHI | 0.09/0.55 | 0.34/0.03 | 0.53/<0.01 | 0.55/<0.01 | 0.40/<0.01 |
| STL | 0.24/0.10 | 0.11/0.48 | 0.18/0.24 | 0.28/0.11 | 0.38/<0.01 |
| IND | 0.24/0.08 | 0.40/<0.01 | 0.33/0.02 | 0.43/<0.01 | 0.21/0.14 |
| CMP | 0.42/<0.01 | **0.63/<0.01** | 0.10/0.51 | **0.74/<0.01** | 0.58/<0.01 |
| BON | 0.60/<0.01 | 0.52/<0.01 | 0.41/<0.01 | **0.68/<0.01** | 0.54/<0.01 |

**Table 4.** Pearson's r, the associated levels of significance (P) and slope for simple linear regression of water-soluble
OPv versus $PM_{2.5}$ mass concentration at five sampling sites. Correlations with r > 0.60 are shown in **bold**. All slope values are in *italic*.
(a)  Water-soluble OP

| | | CHI | STL | IND | CMP | BON |
|---|---|---|---|---|---|---|
| $OP^{AA}$ | Pearson's r/P | -0.02/0.89 | 0.33/0.02 | 0.19/0.18 | 0.54/<0.01 | 0.26/0.09 |
| | *Slope (nmol/min/μg)* | *0.000* | *0.005* | *0.004* | *0.031* | *0.007* |
| $OP^{GSH}$ | Pearson's r/P | 0.45/<0.01 | 0.34/0.02 | 0.45/<0.01 | **0.72/<0.01** | 0.38/0.01 |
| | *Slope (nmol/min/μg)* | *0.005* | *0.003* | *0.005* | *0.016* | *0.005* |
| $OP^{OH-SLF}$ | Pearson's r/P | 0.09/0.55 | 0.26/0.08 | 0.37/<0.01 | 0.43/<0.01 | 0.24/0.12 |
| | *Slope (pmol/min/μg)* | *0.041* | *0.107* | *0.128* | *0.277* | *0.165* |
| $OP^{DTT}$ | Pearson's r/P | **0.62/<0.01** | 0.27/0.07 | 0.55/<0.01 | **0.82/<0.01** | **0.63/<0.01** |
| | *Slope (nmol/min/μg)* | *0.013* | *0.005* | *0.013* | *0.020* | *0.015* |
| $OP^{OH-DTT}$ | Pearson's r/P | 0.24/0.12 | 0.60/<0.01 | 0.37/<0.01 | 0.51/<0.01 | 0.45/<0.01 |
| | *Slope (pmol/min/μg)* | *0.043* | *0.062* | *0.051* | *0.048* | *0.052* |






(b) Methanol-soluble OP

|  |  | CHI | STL | IND | CMP | BON |
|---|---|---|---|---|---|---|
| OP$^{AA}$ | Pearson's r/P | 0.55/<0.01 | 0.12/0.43 | 0.52/<0.01 | **0.64/<0.01** | **0.61/<0.01** |
|  | *Slope (nmol/min/μg)* | *0.010* | *0.002* | *0.010* | *0.011* | *0.012* |
| OP$^{GSH}$ | Pearson's r/P | 0.53/<0.01 | 0.38/<0.01 | 0.51/<0.01 | **0.73/<0.01** | **0.63/<0.01** |
|  | *Slope (nmol/min/μg)* | *0.007* | *0.005* | *0.007* | *0.012* | *0.009* |
| OP$^{OH-SLF}$ | Pearson's r/P | 0.19/0.23 | 0.34/0.02 | 0.45/<0.01 | 0.48/<0.01 | 0.52/<0.01 |
|  | *Slope (pmol/min/μg)* | *0.264* | *0.514* | *0.666* | *0.576* | *0.735* |
| OP$^{DTT}$ | Pearson's r/P | 0.54/<0.01 | 0.49/<0.01 | **0.61/<0.01** | **0.79/<0.01** | **0.61/<0.01** |
|  | *Slope (nmol/min/μg)* | *0.017* | *0.016* | *0.019* | *0.028* | *0.022* |
| OP$^{OH-DTT}$ | Pearson's r/P | 0.25/0.10 | 0.44/0.02 | 0.51/<0.01 | 0.43/<0.01 | 0.50/<0.01 |
|  | *Slope (pmol/min/μg)* | *0.072* | *0.079* | *0.143* | *0.075* | *0.165* |

**Table 5.** Pearson's correlation coefficient (r) and the associated level of significance (P) among various endpoints of
OPv measured at five sampling sites. The values below the diagonal are for water-soluble OPv, while above are for
methanol-soluble OPv. Correlations with r > 0.60 are shown in **bold**.
(a) CHI

| OP endpoint | Pearson's r/significance level (P) for OP endpoints | | | | |
|---|---|---|---|---|---|
|  | OP$^{AA}$ | OP$^{GSH}$ | OP$^{OH-SLF}$ | OP$^{DTT}$ | OP$^{OH-DTT}$ |
| OP$^{AA}$ |  | **0.66/<0.01** | 0.60/<0.01 | **0.69/<0.01** | 0.49/<0.01 |
| OP$^{GSH}$ | 0.32/0.04 |  | 0.30/0.05 | 0.45/<0.01 | 0.17/0.27 |
| OP$^{OH-SLF}$ | 0.09/0.58 | 0.39/<0.01 |  | 0.53/<0.01 | **0.82/<0.01** |
| OP$^{DTT}$ | 0.05/0.73 | 0.40/<0.01 | 0.40/<0.01 |  | **0.64/<0.01** |
| OP$^{OH-DTT}$ | 0.03/0.86 | 0.30/0.05 | 0.48/<0.01 | 0.18/0.24 |  |
|  | OP$^{AA}$ | OP$^{GSH}$ | OP$^{OH-SLF}$ | OP$^{DTT}$ | OP$^{OH-DTT}$ |

(b) STL

| OP endpoint | Pearson's r/significance level (P) for OP endpoints | | | | |
|---|---|---|---|---|---|
|  | OP$^{AA}$ | OP$^{GSH}$ | OP$^{OH-SLF}$ | OP$^{DTT}$ | OP$^{OH-DTT}$ |
| OP$^{AA}$ |  | 0.40/<0.01 | 0.19/0.20 | 0.50/<0.01 | 0.33/0.02 |
| OP$^{GSH}$ | 0.30/0.05 |  | 0.13/0.40 | 0.36/0.01 | 0.23/0.12 |
| OP$^{OH-SLF}$ | 0.51/<0.01 | 0.17/0.26 |  | 0.17/0.26 | 0.42/<0.01 |
| OP$^{DTT}$ | 0.28/0.06 | 0.29/0.05 | 0.22/0.14 |  | 0.57/<0.01 |
| OP$^{OH-DTT}$ | 0.40/<0.01 | 0.38/<0.01 | 0.53/<0.01 | 0.34/0.02 |  |
|  | OP$^{AA}$ | OP$^{GSH}$ | OP$^{OH-SLF}$ | OP$^{DTT}$ | OP$^{OH-DTT}$ |

(c) IND

| OP endpoint | Pearson's r/significance level (P) for OP endpoints | | | | |
|---|---|---|---|---|---|
|  | OP$^{AA}$ | OP$^{GSH}$ | OP$^{OH-SLF}$ | OP$^{DTT}$ | OP$^{OH-DTT}$ |
| OP$^{AA}$ |  | 0.57/<0.01 | 0.54/<0.01 | **0.62/<0.01** | 0.57/<0.01 |
| OP$^{GSH}$ | 0.37/<0.01 |  | 0.59/<0.01 | 0.52/<0.01 | 0.55/<0.01 |
| OP$^{OH-SLF}$ | 0.32/0.02 | 0.23/0.10 |  | 0.44/<0.01 | **0.84/<0.01** |
| OP$^{DTT}$ | 0.17/0.22 | 0.42/<0.01 | 0.44/<0.01 |  | 0.54/<0.01 |
| OP$^{OH-DTT}$ | 0.08/0.58 | 0.20/0.14 | 0.29/0.03 | 0.15/0.29 |  |
|  | OP$^{AA}$ | OP$^{GSH}$ | OP$^{OH-SLF}$ | OP$^{DTT}$ | OP$^{OH-DTT}$ |






(d) CMP

| OP endpoint | Pearson's r/significance level (P) for OP endpoints | | | | |
|---|---|---|---|---|---|
| | $OP^{AA}$ | $OP^{GSH}$ | $OP^{OH-SLF}$ | $OP^{DTT}$ | $OP^{OH-DTT}$ |
| $OP^{AA}$ | | 0.55/<0.01 | 0.46/<0.01 | **0.70/<0.01** | 0.45/<0.01 |
| $OP^{GSH}$ | **0.68/<0.01** | | 0.30/0.04 | **0.69/<0.01** | 0.15/0.32 |
| $OP^{OH-SLF}$ | **0.77/<0.01** | **0.80/<0.01** | | 0.37/<0.01 | **0.66/<0.01** |
| $OP^{DTT}$ | **0.80/<0.01** | **0.73/<0.01** | 0.58/<0.01 | | 0.35/0.01 |
| $OP^{OH-DTT}$ | 0.02/0.91 | 0.26/0.07 | 0.15/0.31 | 0.29/0.04 | |
| | $OP^{AA}$ | $OP^{GSH}$ | $OP^{OH-SLF}$ | $OP^{DTT}$ | $OP^{OH-DTT}$ |

(e) BON

| OP endpoint | Pearson's r/significance level (P) for OP endpoints | | | | |
|---|---|---|---|---|---|
| | $OP^{AA}$ | $OP^{GSH}$ | $OP^{OH-SLF}$ | $OP^{DTT}$ | $OP^{OH-DTT}$ |
| $OP^{AA}$ | | **0.66/<0.01** | **0.77/<0.01** | **0.70/<0.01** | **0.61/<0.01** |
| $OP^{GSH}$ | **0.85/<0.01** | | **0.68/<0.01** | 0.60/<0.01 | 0.53/<0.01 |
| $OP^{OH-SLF}$ | 0.57/<0.01 | **0.64/<0.01** | | **0.69/<0.01** | **0.78/<0.01** |
| $OP^{DTT}$ | 0.51/<0.01 | 0.57/<0.01 | 0.30/0.05 | | **0.68/<0.01** |
| $OP^{OH-DTT}$ | 0.19/0.21 | 0.31/0.04 | 0.28/0.06 | 0.32/0.03 | |
| | $OP^{AA}$ | $OP^{GSH}$ | $OP^{OH-SLF}$ | $OP^{DTT}$ | $OP^{OH-DTT}$ |






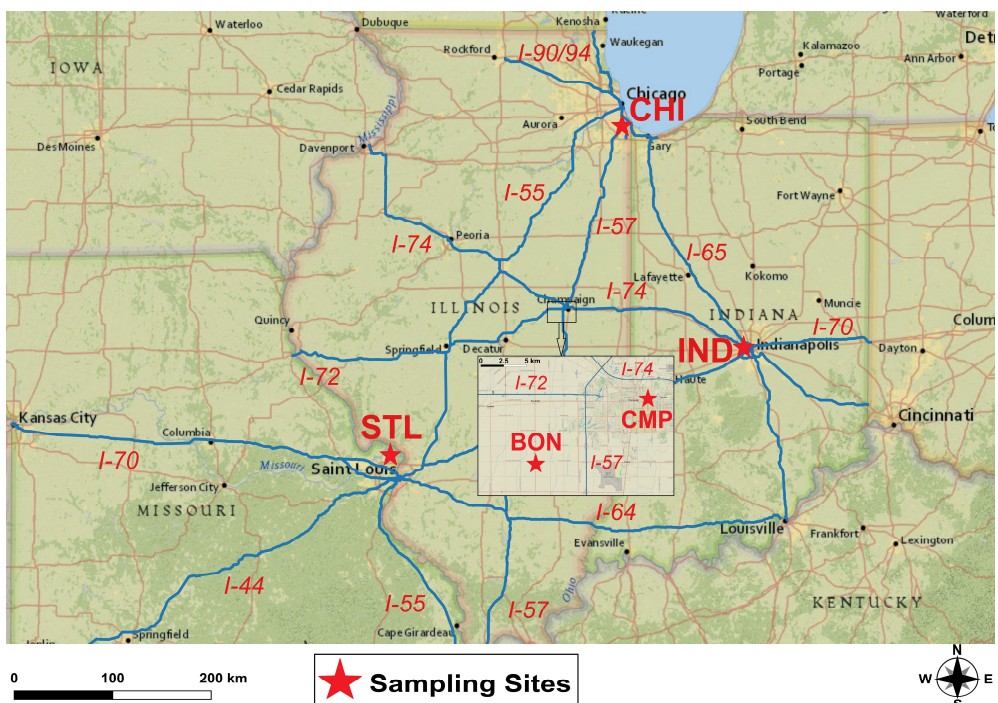


**Figure 1.** Map for our five sampling sites in the Midwest US.

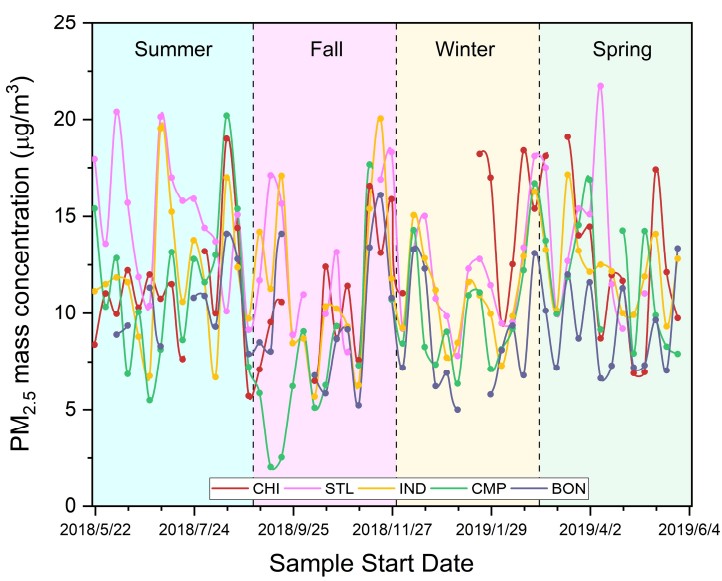


**Figure 2.** Time series of PM$_{2.5}$ mass concentrations at our sampling sites in the Midwest US.



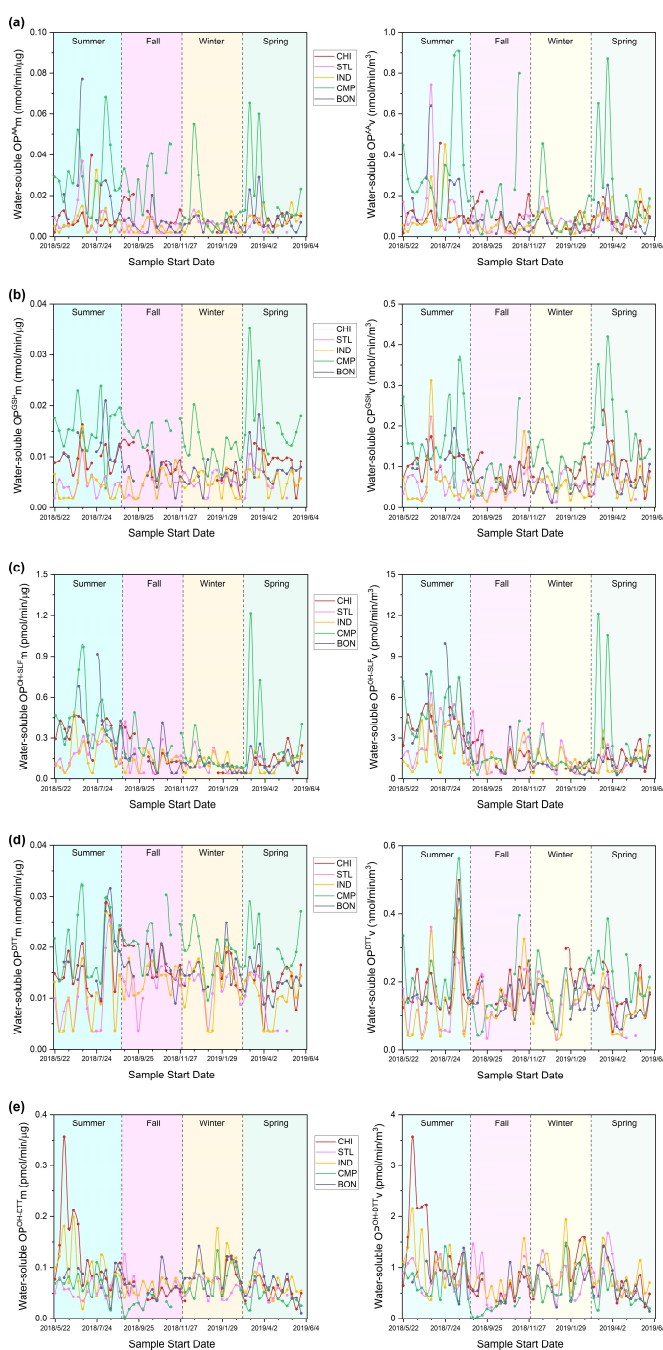


**Figure 3.** Time series of mass-(left) and volume-(right)normalized water-soluble OP activities for (a)
$OP^{AA}$, (b) $OP^{GSH}$, (c) $OP^{OH-SLF}$, (d) $OP^{DTT}$ and (e) $OP^{OH-DTT}$ at our sampling sites.



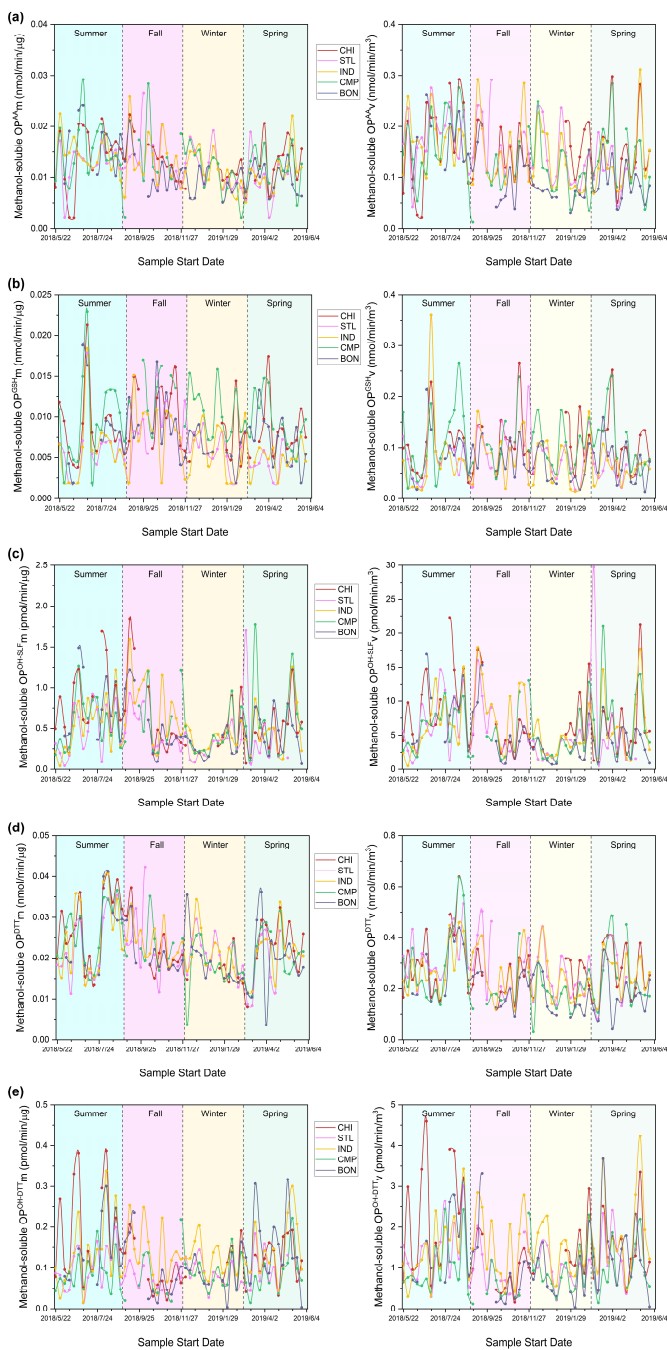


**Figure 4.** Time series of mass-(left) and volume-(right)normalized methanol-soluble OP activities for (a) $OP^{AA}$, (b) $OP^{GSH}$, (c) $OP^{OH-SLF}$, (d) $OP^{DTT}$ and (e) $OP^{OH-DTT}$ at our sampling sites.



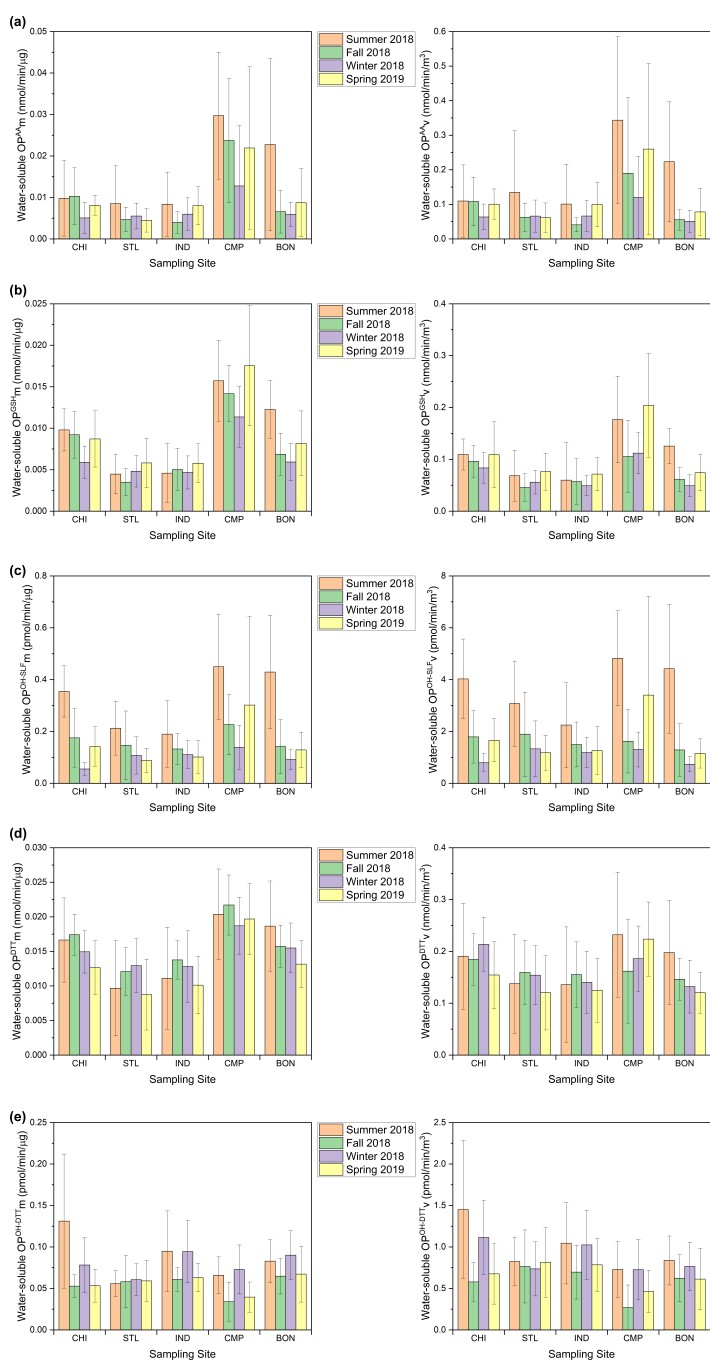


**Figure 5.** Seasonal averages of mass-(left) and volume-(right) normalized water-soluble OP activities for (a) $OP^{AA}$, (b) $OP^{GSH}$, (c) $OP^{OH-SLF}$, (d) $OP^{DTT}$ and (e) $OP^{OH-DTT}$ at our sampling sites.

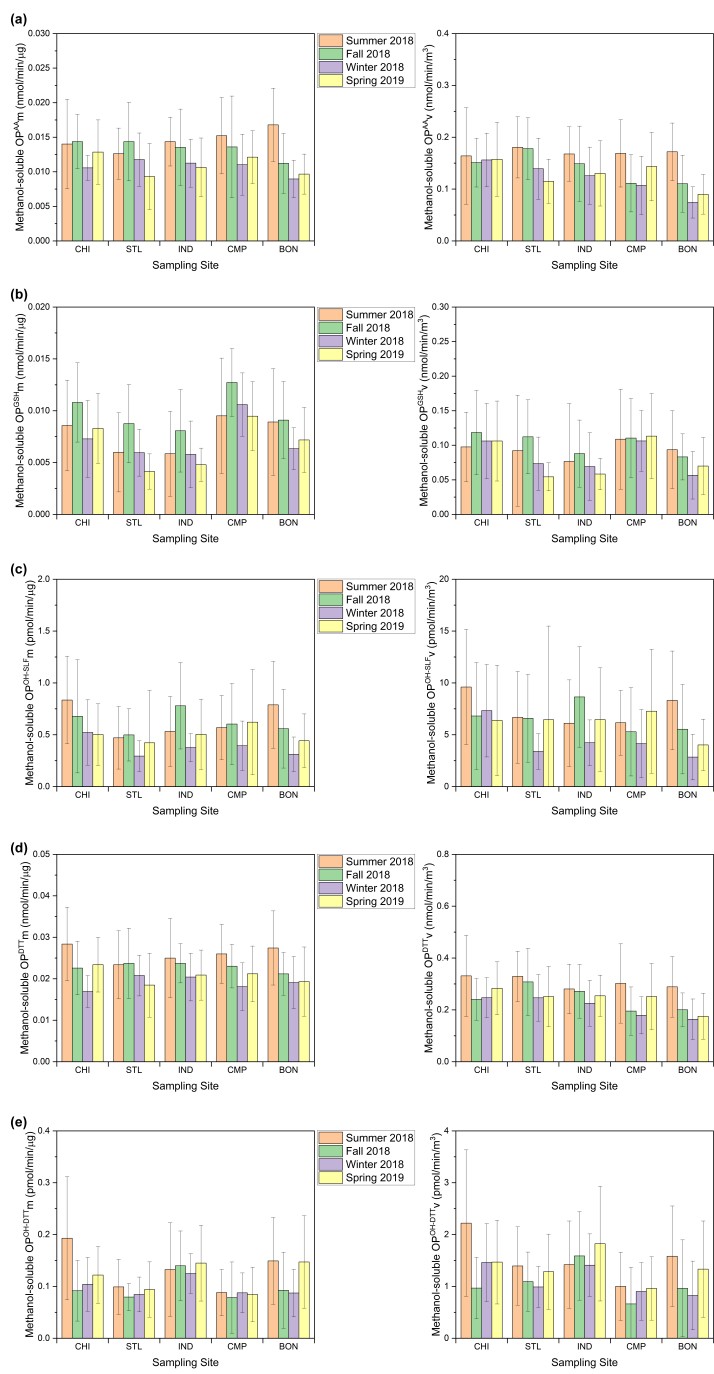


**Figure 6.** Seasonal averages of mass-(left) and volume-(right) normalized methanol-soluble OP activities
for (a) $OP^{AA}$, (b) $OP^{GSH}$, (c) $OP^{OH-SLF}$, (d) $OP^{DTT}$ and (e) $OP^{OH-DTT}$ at our sampling sites.

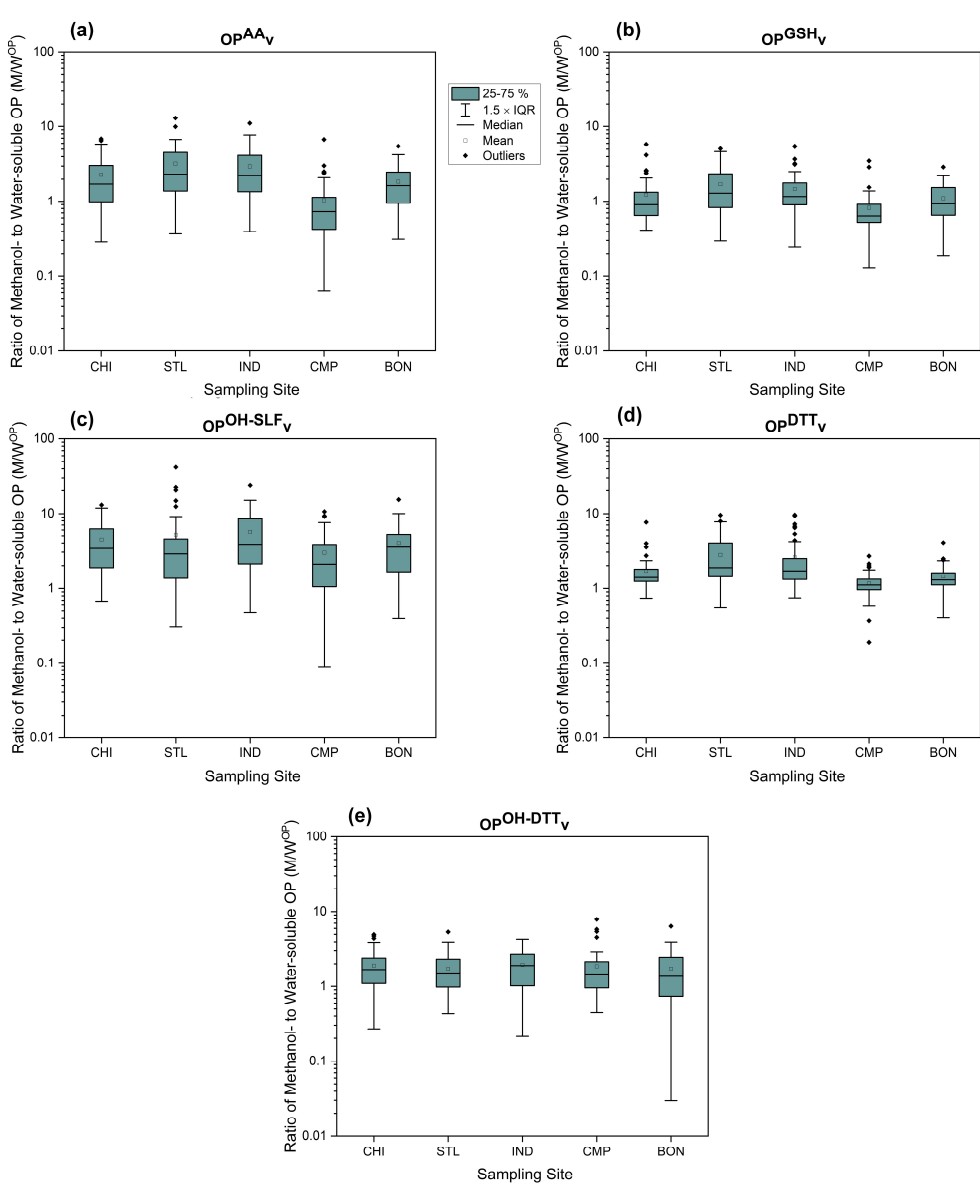

**Figure 7.** Ratio of methanol-soluble OPv to water-soluble OPv (M/W$^{OP}$) for (a) OP$^{AA}$v, (b) OP$^{GSH}$v, (c) OP$^{OH-SLF}$v, (d) OP$^{DTT}$v, and (e) OP$^{OH-DTT}$v at five sampling sites.

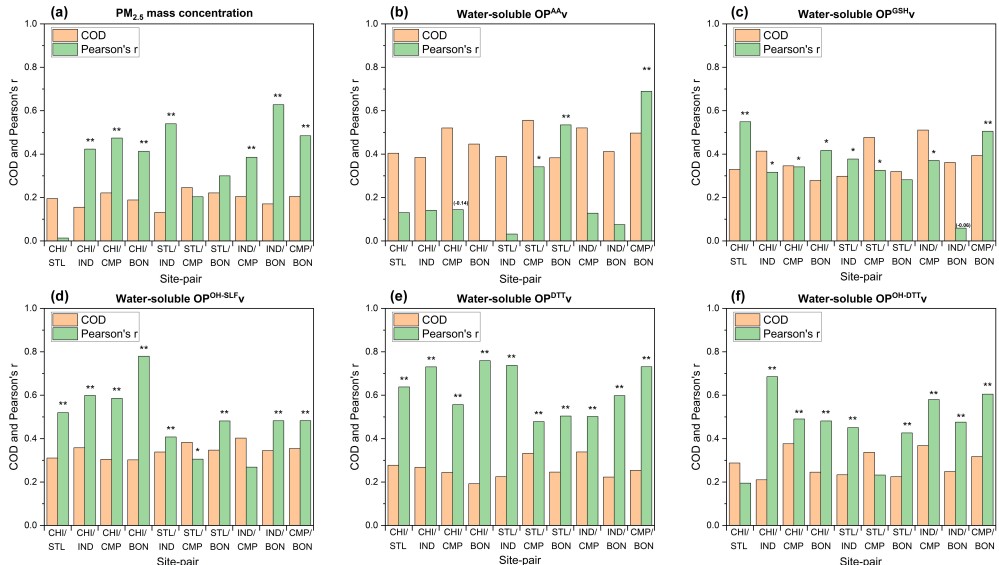

834

**Figure 8.** Coefficient of divergence (CoD) and Pearson's r for site-to-site comparison of (a) PM$_{2.5}$ mass and water-soluble OP activities: (b) OP$^{AA}$v, (c) OP$^{GSH}$v, (d) OP$^{OH-SLF}$v, (e) OP$^{DTT}$v and (f) OP$^{OH-DTT}$v. Asterisks - * and ** on the bars of Pearson's r indicate significant (P < 0.05) and very significant (P < 0.01) correlations, respectively. Note: r for the correlations of OP$^{AA}$v between CHI and CMP and for the correlations of OP$^{GSH}$v between IND and BON were negative (-0.14 and -0.06, respectively).

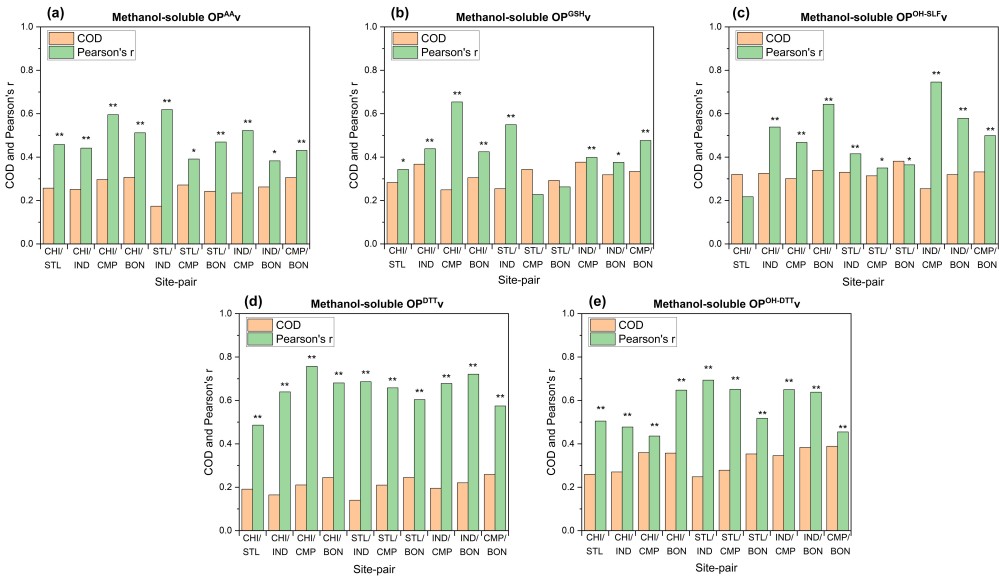

840

**Figure 9.** Coefficient of divergence (CoD) and Pearson's r for site-to-site comparison of methanol-soluble OP activities: (a) OP$^{AA}$v, (b) OP$^{GSH}$v, (c) OP$^{OH-SLF}$v, (d) OP$^{DTT}$v and (e) OP$^{OH-DTT}$v. Asterisks - * and ** on the bars of Pearson's r indicate significant (P < 0.05) and very significant (P < 0.01) correlations, respectively.