# Peer review of "Spatiotemporal Variability in the Oxidative Potential of Ambient"

_Atmospheric Chemistry and Physics, 2021_

## Author Comment (AC1)

Manuscript ID: acp-2021-376                                                08/23/2021

Dr. Timothy Bertram
Editor
Atmospheric Chemistry and Physics

Dear Dr. Timothy Bertram,

Along with this letter, we have submitted our response document for the manuscript "Spatiotemporal Variability in the Oxidative Potential of Ambient Fine Particulate Matter in Midwestern United States". We had received the reviews from two referees and one community comment. All the comments have been satisfactorily addressed based on a point-by-point response in the attached document. To facilitate the review process, we have also included the marked-up version of our revised manuscript (track-changes mode), so that the reviewers can see how the comments are incorporated in the manuscript. The manuscript has been substantially improved as a result of this review and we really appreciate all the valuable suggestions provided by the reviewers.

We believe that our revised manuscript meets the high-quality standards of ACP, and we look forward to any further comments the reviewers and editor might have.

Sincerely,

Haoran Yu

Graduate Student
Department of Civil and Environmental Engineering
University of Illinois at Urbana-Champaign
N Mathews Ave, Urbana, IL 61801
352-213-5899

**Reviewer: Anonymous Reviewer #2**

Yu et al report on extensive measurements of PM2.5 OP (oxidative potential) based on an analysis involving 5 different acellular approaches. The analysis was performed on samples collected at a number of sites in the midwestern US and the paper reports on comparisons between the assays and PM2.5 mass. It is stated that a second paper will focus on the PM2.5 chemical components driving these results. The paper is based on a substantial amount of work and provides more insights into the utility of current ways to characterize OP, and it also sheds light on the potential usefulness of using these assays in health studies.

A major conclusion is that the poor correlation between all the various assays, when compared at one site, (and this is largely true for all the sites), implies all these types of OP assays are needed for health studies. One could also conclude, that all of these assays (except possibly one) are each deficient, and no ideal assay exists. It may also even suggest that if no comprehensive OP assay is available, then maybe the approach is flawed since the goal of using these assays was to develop a comprehensive single measure of aerosol toxicity. Since this group of assays appears to fail in demonstrating this goal, instead maybe one should focus on the specific species that drive OP and not use these assays? How does one know if even more assays are needed to fully characterize PM2.5 OP? Furthermore, how would all these various OP measurements, even if available to health researchers, be utilized in a health study, ie how would they be combined to give an overall better indicator of PM2.5 OP? These questions are important and should likely be considered; a discussion beyond the conclusion that all these assays should be utilized, is warranted.

*Response:*

*We thank the reviewer for the inspiring comments. These comments have really helped us in enhancing the discussion of our paper. The reviewer raised several questions regarding the rationality of using oxidative potential (OP) as a health indicator and measuring OP with multiple endpoints. We have attempted to address them point-by-point in the following discussion.*

"One could also conclude, that all of these assays (except possibly one) are each deficient, and no ideal assay exists."

*Yes, we agree with the reviewer's comment that one aspect of the conclusion of our study could be that all of these assays are each deficient, and no ideal assay exists. However, to be more accurate, we cannot comment on the deficiency or benefit of an assay based on this study. This will require an integration of these assays with either toxicological or epidemiological study. Nevertheless, following the reviewer's suggestion we have added a few sentences in the results and discussion section of our manuscript in lines 576 – 585, "Overall, a poor-to-moderate and inconstant intercorrelation trend among different endpoints of both water-soluble and methanol-soluble OP at most sites indicates that all these assays could be deficient from being ideal and measuring a single endpoint is not enough to represent the overall OP activity. ... However, it should be noted that our study is not designed to assess and rank the biological relevance of these acellular endpoints, which will require an integration of these and possibly other novel assays involving different routes of oxidative stress, in either toxicological or epidemiological studies." We also*

*included it in our conclusion in lines 613 - 615, "Since our study cannot comment on the biological relevance of these different pathways, we recommend integrating all these and other assays in toxicological or epidemiological studies, to assess their relative utilities."*

"It may also even suggest that if no comprehensive OP assay is available, then maybe the approach is flawed since the goal of using these assays was to develop a comprehensive single measure of aerosol toxicity."

*We do not agree with the reviewer's point here. First, we do not think that the goal of these assays was to develop a comprehensive single measure of aerosol toxicity. The current national ambient air quality standards are based on PM mass alone, despite we clearly know that certain components of the PM are more toxic than others. One goal of developing these assays was to have an alternative metric which is able to capture some of the potential toxic mechanisms of these components. Although it could appear from the OP literature that the goal is to develop a single measurement of OP for representing multiple pathways of aerosol toxicity, numerous studies have repeatedly indicated that all these measures have their limitations in terms of incorporating the roles of different redox-active components. For example, Xiong et al. (2017) reported negligible $OP^{DTT}$ activity of Fe ions (i.e. $Fe^{2+}$ and $Fe^{3+}$) and strong synergistic effect of Fe and quinones in $OP^{OH-DTT}$, indicating the limitation of $OP^{DTT}$ in counting the contribution of Fe. Ayres et al. (2008) reported different responses of $Fe^{3+}$, $Cu^{2+}$ and $Zn^{2+}$ towards $OP^{AA}$ and $OP^{GSH}$ in a respiratory tract lining fluid (RTLF). Moreover, many studies have found different correlation trends of different endpoints with chemical components and sources of PM, e.g. $OP^{AA}$ vs. $OP^{DTT}$ (Fang et al., 2016;Perrone et al., 2019;Visentin et al., 2016;Janssen et al., 2014), $OP^{ESR}$ (i.e. oxidative potential measured with electron spin resonance assay) vs. $OP^{AA}$, $OP^{GSH}$ and $OP^{DTT}$ (Calas et al., 2018). Janssen et al. (2015), Weichenthal et al. (2016a), Weichenthal et al. (2016b) and Maikawa et al. (2016) also reported different associations of different acellular OP endpoints (e.g. $OP^{AA}$, $OP^{GSH}$, $OP^{DTT}$ and $OP^{ESR}$) with the health endpoints, including markers of airway and nasal inflammation, risk of emergency room visits for respiratory diseases, myocardial infarction, and fractional exhaled nitric oxide (FeNO), respectively. However, despite these differences and limitations, we do not think that it is appropriate to say that the approach is flawed, simply because in almost all of the health studies, these assays have shown a better association than the PM mass (Bates et al., 2015;He et al., 2021;Maikawa et al., 2016;Strak et al., 2017;Weichenthal et al., 2016a). Thus, we know that despite their limitations they are superior to the currently used PM metric based solely on the mass. These evidences show the complexity of OP-associated pathways, and make it somewhat unrealistic to develop a single comprehensive assay, at least with the current state of the art.*

*Given the current scenario, it sounds reasonable to combine these assays, i.e. apply all of these assays on each PM sample, for assessing the OP comprehensively. Although each assay has its deficiency, it can represent a specific pathway of OP which probably overcomes the deficiency of another assay lacking that particular pathway. For example, $OP^{OH-DTT}$ developed in our previous studies (Xiong et al., 2017;Yu et al., 2018) can supplement the pathway represented by $OP^{DTT}$ for generating superoxide radical ($\cdot O_2^-$), with its subsequent reaction with metal ions for generating the hydroxyl radical ($\cdot OH$). $OP^{AA}$ and $OP^{GSH}$ directly measure the consumption of these antioxidants (i.e. AA and GSH) in a surrogate lung fluid (SLF), representing the antioxidant consumption pathways, while measuring $\cdot OH$ generation in SLF ($OP^{OH-SLF}$) simulate subsequent*

*reactive oxygen species (ROS) generation process in human lung lining fluid and thus supplementing the antioxidant consumption process. These five assays combined together cover most of the known and potentially important biological pathways of PM exerting oxidative stress in vivo. Our results showing disparities in the intercorrelation among five endpoints further support the finding that by combining these five assays, we can minimize their deficiencies.*

"Since this group of assays appears to fail in demonstrating this goal, instead maybe one should focus on the specific species that drive OP and not use these assays?"

*Measuring the specific species in PM that drive OP is even more complicated in linking the chemical composition with health effects. First, the composition of PM is highly complex containing tens of trace metals (Kundu and Stone, 2014;Kim et al., 2005;Luo et al., 2018;Reff et al., 2009;Tao et al., 2017), innumerous organic species (Lin et al., 2017;Lin et al., 2018;Lin and Yu, 2020;Riva et al., 2016;Chen et al., 2020) and numerous inorganic ions ($NH_4^+$, $SO_4^{-2}$, $NO_3^-$, etc.). Note, none of the analytical techniques is capable of measuring all of the organic compounds, therefore bulk parameters such as OC, WSOC and humic-like substances (HULIS) are used to represent such a large group of species present in the ambient PM. Despite such classifications, these bulk organic species coming from different sources show very different OP behavior. For example, Lin and Yu (2020) reported three different types of interactions, i.e. additive, antagonistic, and synergistic of the HULIS extracted from three different sources, i.e. ambient $PM_{2.5}$, rice straw burning and sugar cane leaf burning, respectively, with Cu for oxidizing AA. Second, the health effect of PM might not be accounted by simply adding up the contribution of individual chemical species due to non-linear responses of some species like Cu and Mn towards OP (Charrier and Anastasio, 2012;Charrier et al., 2015) and synergistic/antagonistic interactions among various PM species for exerting the oxidative stress and toxicity (Lin and Yu, 2020;Yu et al., 2018;Charrier and Anastasio, 2015;Wang et al., 2020). All these points essentially demonstrate that the approach of relating the health effects directly with the chemical composition is even more complicated than using rather limited number of the OP assays.*

"How does one know if even more assays are needed to fully characterize PM2.5 OP?"

*We completely agree with the reviewer on this point. There could be more assays needed to fully characterize the $PM_{2.5}$ OP. This is an open question which we do not think can be addressed from our study and neither it was the goal of the current analysis. However, as of now, these are the most commonly used endpoints, all of which we have included in our study. As the knowledge on this topic expands, we expect that future investigations on the novel OP endpoints might extend our scope. Following the reviewer's suggestion, we have included this point in the discussion of our manuscript in lines 578 – 583, "Although, the OP endpoints used in our study have covered some of the well-known and important pathways of the in vivo oxidative stress caused by $PM_{2.5}$, there are other endpoints (e.g. consumption of cysteine, formation of $H_2O_2$, etc.), and more assays can be developed in the future. We suggest that a collection of diverse range of OP endpoints, measured separately as done in our study could better capture the role of different PM components and their interactions via different pathways for driving the oxidative levels of the PM in a region."*

"Furthermore, how would all these various OP measurements, even if available to health researchers, be utilized in a health study, ie how would they be combined to give an overall better indicator of PM2.5 OP?"

*First, we would like to highlight that the importance of our study lies in showing that the responses of these assays do not correlate with each other. Which of these assays is better than the other is the second question which is beyond the scope of our current study. To address that question, we need to integrate them in the epidemiological studies. However before that step, an obvious question arises that do all these assays have to be integrated or just few of them (in case they would have been correlated). Our investigation shows that all of them should be integrated to know which one is better than the other, because they are not correlated with each other.*

*Now, by combining, we do not mean to merge them into one assay, rather we mean that we should do all of them individually on each PM sample. Then we should integrate all of this data in an epidemiological study to assess the relevance of each of them. Some previous studies have adopted this approach for investigating the health relevance of OP by associating it with health endpoints (Abrams et al., 2017;Strak et al., 2017;Zhang et al., 2016;Yang et al., 2016;Weichenthal et al., 2016a;He et al., 2021;Janssen et al., 2015). These studies have definitely helped in enhancing our understanding on the relevance of OP measurements and the role of specific endpoint in comparison to PM mass. However, these are very limited with their focus only on 2 or 3 endpoints. Incorporating all the available OP endpoints measured on the same set of samples in epidemiological studies should help to clearly see their roles and rank them as per their relevance, which is what we expect in longer term from this dataset.*

The data do support other studies showing variability between various OP measures and PM2.5 mass, suggesting PM2.5 mass is a poor predictor of the ability of particles to cause oxidative stress (assuming these assays are good measures of OP). This is an important finding.

*Response:*

*We thank the reviewer for appreciating this finding.*

Comparisons between sites using different samplers operating at the same time depends on some level of measurement precision to argue that observed differences (poor correlations) are really due to differences in aerosol particles at the sites. This applies to the gravimetric measurement of PM2.5 mass and the various OP measurements. The authors do discuss variability in the negative and positive controls, but the data shown in Table 1 is only the precision of the analysis and does not consider sampling, filter storage or extraction. Can it be stated that this precession for all the species measured and PM2.5 mass is significantly better (lower variability) than that of the comparisons between sites. It would be especially interesting to know the precession of the methanol extracts, which based on the extraction approach is likely the most imprecise measurement (curiously it also shows the least variability between assay results from various sites). A more comprehensive discussion is warranted that includes specifically addressing if the differences seen are real or just noise.

*Response:*

*This is a good point by the reviewer and we apologize not to address it earlier in our original manuscript, despite conducting some experiments to test the variability among various samplers, before the sampling. To further explore it, we have conducted more experiments now after the sampling. The results of all these experiments are presented in the discussion below.*

*First of all, we would like to note that out of five samplers used in our study, two were old samplers (about 5 years old, used in various sampling campaigns) and three were brand new, which were bought from TISCH Environmental (Cleves, OH, US) a month before the sampling. These new samplers were factory calibrated and installed at three farther sites, i.e. Chicago (CHI), Indianapolis (IND) and St. Louis (STL). The other two old samplers were installed at Champaign (CMP) and Bondville (BON). For the sole purpose of this discussion, we will name them as CHI (N), IND (N), STL (N), CMP (O) and BON (O). Since the new samplers were factory calibrated, we had more confidence in them, therefore, we chose one of those samplers, i.e. CHI(N), as a reference and compared the responses of other two old samplers, i.e. CMP (O) and BON (O), by running them in pairs, i.e. first CHI (N) and CMP (O) pair, followed by CHI (N) and BON (O) pair, at a site in Urbana in April 2018 (due to some practical constraint, we couldn't run all three of them together). We collected 9 sets of Hi-Vol samples on the quartz filters (24-hours integrated samples) from each pair, and analyzed them for the DTT assay using the same extraction and analysis procedure as used in our current study. The comparison of this analysis is shown in Figure 1 of the response document. As can be seen from these figures, there are excellent correlations ($R^2 = 0.92 - 0.94$) between the old and new samplers, with slopes almost equal to 1.*

[Figure]

*Figure 1. Comparison between $OP^{DTT}$ of the $PM_{2.5}$ samples collected from three samplers: CHI (N) vs. CMP (O) (Figure 1a) and CHI (N) vs. BON (O) (Figure 1b)*

*After this comparison, we moved all the samplers to their respective sites for the campaign. We believe, that the largest cause of uncertainty in these samplers when they were moved to different sites should be from the variability in their flow rates. Therefore, to minimize that, we always measured the flow rates before and after collecting the $PM_{2.5}$ samples. During the entire sampling campaign, all five samplers were monthly calibrated for the flow rate by using a variable flow calibration kit (Tisch Environmental), which includes a calibration orifice and slack tube water manometer.*

*We controlled the variability from gravimetric measurements by weighing the filters for at least three times before and after sampling, and ensured that the maximum difference of the mass between three consecutive weighing was less than 0.5 mg. This value is insignificant in comparison to the typical PM$_{2.5}$ mass loadings on the filters, i.e. 40 – 100 mg. Moreover, we always stored all our samples in the same freezer at -20 °C right after weighing. The samples were only taken out from the freezer prior to OP analysis and were immediately placed in the freezer after punching to minimize the loss of semi-volatile species. This should eliminate the effect of storage on the precision.*

*However, we understand that despite these quality control and checks, we should still inter-compare the three new Hi-Vol samplers installed in Chicago, Indianapolis and St. Louis. Therefore, following the reviewer's comment, we brought these samplers back to our university last month, put them side-by-side at a site in Urbana (IL) and collected 9 Hi-Vol samples (24-hour integrated) from each sampler. All these samples were extracted and analyzed for the DTT activity in the same manner as used in our current study. The results of these comparisons are shown in Figure 2 of the response document. Again, we found excellent correlations ($R^2 = 0.93 – 0.95$) with slopes close to 1. Note, these comparison results include the variabilities caused by sampling, filters storage and their extraction, as pointed out by the reviewer.*

[Figure]

*Figure 2. Comparison between OP$^{DTT}$ of the PM$_{2.5}$ samples collected from three samplers: CHI (N) vs. STL (N) (Figure 2a) and CHI (N) vs. IND (N) (Figure 2b)*

*Finally, to address the reviewer's comment related to methanol extracts, we assessed the precision of methanol-soluble OP for all endpoints, following the same protocol as used for the water-soluble OP measured in our previous study (Yu et al. (2020)). Specifically, ten groups of four punches, each of 0.75'' diameter were cut from the same Hi-Vol filter collected at CMP site, and extracted separately into 10 mL methanol. The methanol in the filtered extracts was then evaporated, and each individual residual extract (~50 µL) was reconstituted with DI to reach 12 mL volume. The concentration of the PM in the reaction vial (RV) was maintained at the same level as used in Yu et al. (2020), i.e. 50 µg/mL for SLF-based endpoints, and 30 µg/mL for DTT-based endpoints. The coefficient of variation (CoV; i.e. the standard deviation of the ten groups of measured OP divided by their average), was used to determine the precision of OP and shown in Table 1 of this response document. Overall, the CoV for methanol-soluble OP of all endpoints (8.9*

*– 14.5 %) was at the same level as that for the water-soluble OP (7.9 – 13.3 %) reported in Yu et al. (2020), indicating that the precision of methanol-soluble OP was as good as water-soluble OP. We have included all these results in SI (Section S1, Figures S1-S2 and Table S2) of the revised manuscript, and discussed them in lines 141 – 142 of the revised manuscript, "Both before and after the sampling campaign, we did a comparison of various samplers by running them in parallel to collect PM$_{2.5}$ samples and analyzing them for OP$^{DTT}$ (see Section S1 of the supplemental information, SI). ... All five samplers were monthly calibrated for the flow rate by using a variable flow calibration kit (Tisch Environmental), and the flow rate was measured every week before and after the sampling.", and lines 228 – 234, "The precision of SAMERA was assessed previously using water-soluble extracts and the coefficient of variations (CoVs) were reported to be less than 14 % (7.9 – 13.3 %) for all OP endpoints (Yu et al., 2020). We also assessed the precision using methanol-soluble extracts and found similar levels of CoVs, i.e. 8.9 -14.5 % for all OP endpoints (see Table S2 in SI). Consistency of our current results for negative controls with those reported earlier, and the low CoVs obtained for the positive controls (1.1 – 11.8%), and PM$_{2.5}$ extracts ensured a good quality assurance for the overall OP analysis."*

*Table 1. Precision of SAMERA for methanol-soluble OP measurements compared with water-soluble OP measurements.*

| Endpoint | Unit | Average | Standard Deviation | CoV (%) | CoV (%) for the water-soluble PM$_{2.5}$ extract (Yu et al., 2020) |
|---|---|---|---|---|---|
| OP$^{AA}$ | nmol/min/m$^3$ | 0.132 | 0.018 | 13.51 | 11.87 |
| OP$^{GSH}$ | nmol/min/m$^3$ | 0.098 | 0.010 | 10.65 | 7.89 |
| OP$^{OH-SLF}$ | pmol/min/m$^3$ | 0.740 | 0.011 | 14.49 | 10.56 |
| OP$^{DTT}$ | nmol/min/m$^3$ | 0.187 | 0.017 | 8.89 | 10.52 |
| OP$^{OH-DTT}$ | pmol/min/m$^3$ | 0.216 | 0.023 | 10.88 | 13.28 |

One conclusion that may be drawn from this work and which is consistent with past studies is that the DTT assay is the most comprehensive measurement of OP (see, for example, discussion in lines 289-407). This may be because DTT includes electron transfer reactions from both organic species and metals, whereas AA, GSH and production of OH in the various assays is likely largely driven by metals. One could actually discuss an interpretation of the data in which the most assay meets the goal of being the most comprehensive. For example, maybe instead of arguing that all assays in their various forms are needed, one could try to assess which is best?

*Response:*

*We agree that OP$^{DTT}$ has been widely used in many studies as the OP indicator, and it was associated with both organic species (e.g., HULIS, quinones) and metals (e.g., Cu and Mn) (Charrier and Anastasio, 2012;Yu et al., 2018). However, as we have pointed out earlier, OP$^{DTT}$ does not capture the contribution of Fe in ·OH formation (Xiong et al., 2017;Yu et al., 2018). This mechanism of ROS generation is also important as shown in one of our earlier study revealing the synergistic interaction of Fe with quinones and HULIS in enhancing the cytotoxicity (Wang et al., 2020). As observed in many studies, this synergism between Fe and organic species was captured by both OP$^{OH-SLF}$ (Wei et al., 2018;Gonzalez et al., 2017) and OP$^{OH-DTT}$ (Yu et al., 2018;Xiong et al., 2017). Wang et al. (2018) reported stronger correlations of cytotoxicity of ambient PM$_{2.5}$ with*

*both $OP^{OH-SLF}$ and $OP^{OH-DTT}$ (r = -0.84 and -0.82, respectively) compared to its correlation with $OP^{DTT}$ (r = -0.58), further indicating that both ·OH generating endpoints could have more important roles in the biological pathways leading to cytotoxicity. Similarly, although $OP^{AA}$ and $OP^{GSH}$ showed similar sensitivities as $OP^{DTT}$ towards certain species (i.e. Cu), they represent potentially different biological pathways of oxidative stress. $OP^{DTT}$ simulates the redox reaction of cellular antioxidants, such as NADPH in mitochondria (Cho et al., 2005;Kumagai et al., 2002), while $OP^{AA}$ and $OP^{GSH}$ directly measure the antioxidant consumption in lung lining fluid (Weichenthal et al., 2016b). Previous studies have also noted some associations of health outcomes with $OP^{AA}$ (Janssen et al., 2015) and $OP^{GSH}$ (Maikawa et al., 2016;Weichenthal et al., 2016b), respectively.*

*Considering the deficiencies and biological relevance of each endpoint, we believe it would be premature to rank $OP^{DTT}$ as the best assay among them. Rather than the comparison among themselves or their correlation with the chemical composition, we think that the choice of the most comprehensive OP endpoints (if there is any such thing) should be determined by their association with the health outcomes.*

Specific Comments.

Line 20-21, not sure how higher site to site correlations proves methanol extracts includes more insoluble species? The idea that methanol extracts a greater fraction of OP than water is well known.

*Response:*

*Water-extracts are supposed to contain only water-soluble components while methanol being a solvent with polarity between water and strongly non-polar solvents such as hexane, is supposed to extract major fraction of both water-soluble and water-insoluble components. Our rationale for explaining higher site-to-site correlation in methanol extracts is that the components coming from same sources, such as the regional sources (SOA, biomass burning etc.) have a better chance of being extracted in methanol (irrespective of whether they are water-soluble or insoluble) and thus lead to a higher correlation, masking the effect of the components originated from local sources which could have a narrow range of solubilities. We have further clarified it in our sentences on lines 532 – 536, "It is possible that methanol is able to extract more redox-active PM components coming from regional emission sources, e.g. biomass burning or secondary organic aerosols, present at these sites. The components originated from these common sources could mask the effect of other components originated from the local sources having a narrower range of solubilities, thus yielding to an overall lower spatiotemporal variability and better correlation among different sites."*

Lines 142 to 148, Charrier et al (2016) suggest a mass concentration for measurement of OP to limit nonlinear effects of 10ug PM/mL, here the authors use 100 ug/mL, why and what is the effect of doing this, ie does it solve the nonlinear problem?

*Response:*

*We clarify that the concentration of PM$_{2.5}$ in the extract we used for measuring OP is 30 µg/mL for OP$^{DTT}$ and OP$^{OH-DTT}$, and 50 µg/mL for OP$^{AA}$, OP$^{GSH}$ and OP$^{OH-SLF}$ (lines 154-156 in the original preprint). The concentration of 100 µg/mL was used in the sample vials kept in our automated system, which were further diluted before using them in the reaction vials. Note, the range recommended by Charrier et al. (2016) was based on the samples collected from California (Claremont and Fresno). OPm is a sole function of PM chemical composition and this recommendation of the standard concentration is not applicable to the samples with different chemical composition. Charrier et al. (2016) also noted that there is no "right" concentration for the standard. As quoted from their publication, "We propose a standard of expressing mass-normalized DTT results relative to an extract concentration of 10 mg-PM/mL of DTT solution; while there is no 'right' concentration for the standard, this proposed extract concentration provides an adequate DTT response for typical ambient PM in our experience but uses relatively little sample." For DTT-based endpoints, our preliminary tests indicated that the concentration recommended in Charrier et al. (2016) (10 µg/mL) was very low for some of our samples with low redox activity, while 30 µg/mL of PM$_{2.5}$ extract was the safe concentration to produce the levels well above detection levels for OP$^{DTT}$ and OP$^{OH-DTT}$ activities. Since our samples are collected from Midwest US, there could be a very different mix of aerosol sources for our samples compared to their (Charrier et al., 2016) samples collected in California. Thus, it is reasonable to choose the concentration based on the specific composition of our samples to obtain effective measurements.*

*We adopted the concentration for SLF-endpoints based on many previous studies using OP$^{AA}$ and OP$^{GSH}$ as the OP indicators (Godri et al., 2011;Godri et al., 2010;Ayres et al., 2008;Künzli et al., 2006;Szigeti et al., 2016). This concentration was sufficient for producing valid OP$^{OH-SLF}$ values (i.e. higher than the detection limit of our measurements) for most of our PM$_{2.5}$ samples.*

*Moreover, since we are keeping the concentration constant across all samples, the non-linear biases caused by the concentration of Cu and Mn in the OP endpoints are not so relevant for the comparison of OP responses of our samples collected from different sites.*

It would be useful to provide the composition of the simulated lung fluid.

*Response:*

*The surrogate lung fluid (SLF) used in our study consists of four antioxidants. The final concentrations of these antioxidants in the reaction vial used for incubating with the PM extract were 200 µM L-ascorbic acid (AA), 100 µM reduced glutathione (GSH), 300 µM citric acid (CA) and 100 µM uric acid (UA). We have included the procedures for making SLF and the final concentrations of these antioxidants in the manuscript in lines 187 – 190 , "SLF was made following the protocol of Yu et al. (2020), i.e. by mixing equal volumes (1 mL each) of four antioxidant stock solutions – 20 mM AA, 10 mM GSH, 30 mM citric acid (CA) and 10 mM UA, and diluting the mixture by DI to 10 mL. Final concentrations of the antioxidants in the RV used for incubating the sample, were 200 µM AA, 100 µM GSH, 300 µM CA and 100 µM UA."*

One issue with current measurements of OP by the various methods is that there is a range of approaches used for each of the methods.  This makes comparisons between this work and other studies complicated. It would be valuable to know exactly how these various methods compare to what has been utilized in other studies. For example, maybe a table in the supplement could provide more details on the methods used here links to past studies that used the exact same approach.

*Response:*

*In Table S2 of our submitted preprint, we have included the studies using the same OP endpoints, and briefly described the differences of their methods in the notes. We thank the reviewer for this suggestion, based on which we have further expanded this table by including more details of the methodology of the studies we cited in the revised Table S6 (corresponding to Table S2 of the preprint).*

Line 238-239, this statement should be supported with data.

*Response:*

*We have conducted one-way analysis of variance (ANOVA) test on both spatial and temporal variability of $PM_{2.5}$ mass. The results are included in SI Table S3, and the P-values are added in lines 268 – 272, "The highest mass concentrations were recorded at CHI during winter ($P < 0.01$; Table S3) and STL during summer ($P < 0.05$), while BON exhibited the lowest concentrations in all seasons, except fall when the mass concentrations were lowest at CMP ($P < 0.05$). Other than these minor variations, the $PM_{2.5}$ mass concentrations are both spatially and temporally homogeneous in the Midwest US with no significant seasonal differences ($P > 0.05$ at most sites)." We also added median values in lines 265 – 268, "Generally, the more urbanized sites of our study (i.e. CHI, STL and IND) showed slightly higher mass concentrations (5.7 – 21.7 $\mu g/m^3$, median: 11.8 $\mu g/m^3$) compared to the smaller cities like CMP and its rural component (i.e. BON) (2.0 – 20.2 $\mu g/m^3$, median: 9.2 $\mu g/m^3$)." to support our statement.*

Line 274, typo, change "into" to "in"?

*Response:*

*We have made this change.*

How do the authors explain the data where OP in water extracts is greater than OP methanol when it is established that methanol extracts water soluble species plus organic species? Seems this result demonstrates the lack of precision of the methanol method. Or are the authors implying that some water soluble species that contribute to OP are not extracted and detected in the methanol method?

*Response:*

*We do not agree with the reviewer on the lack of precision of the method for methanol extraction and analysis. As shown in Table 1 of the response document, the precision of methanol-soluble OP is as good as water-soluble OP.*

*The measured OP of PM is not simply the addition of the activities of all extracted PM components. Previous studies have reported both synergistic and antagonistic interactions among transition metals and organic species in multiple endpoints, such as $OP^{AA}$ (Lin and Yu, 2020, 2021), $OP^{OH-SLF}$ (Gonzalez et al., 2017;Wei et al., 2018;Charrier and Anastasio, 2015), $OP^{DTT}$ (Yu et al., 2018;Xiong et al., 2017) and $OP^{OH-DTT}$ (Yu et al., 2018;Xiong et al., 2017). Hence, lower methanol-soluble OP does not necessarily imply fewer extracted species in methanol. Here, we infer that the lower methanol-soluble $OP^{AA}v$ than water-soluble $OP^{AA}v$ might be attributed to the antagonistic effect from the additional components in methanol-soluble extracts. Lin and Yu (2020) reported an antagonistic interaction between HULIS extracted from rice straws burning and Cu on $OP^{AA}v$. They found an abundance of alkaloid compounds in the HULIS, which can chelate Cu and reduce its reactivity with AA. Although we have not yet conducted chemical composition analysis, it is possible that the $PM_{2.5}$ samples collected at CMP could be strongly impacted by biomass burning sources and therefore could contain high levels of alkaloids. Our previous studies also found an elevated level of Cu [up to 60 $ng/m^3$, compared to the typical Cu concentration (4 – 20 $ng/m^3$) at most urban sites in US (Baumann et al., 2008;Buzcu-Guven et al., 2007;Hammond et al., 2008;Kundu and Stone, 2014;Lee and Hopke, 2006;Milando et al., 2016)] at CMP (Wang et al., 2018;Puthussery et al., 2018). Since many of the alkaloid compounds are methanol-soluble but water-insoluble, it is possible that these compounds are more efficiently extracted in methanol and are complexed with a large fraction of Cu, thus causing lower levels of methanol-soluble $OP^{AA}v$ compared to water-soluble $OP^{AA}v$ at CMP. We have included this inference in lines 346 – 355 of the original preprint and lines 467 – 479 in the revised manuscript.*

What is the difference between methanol soluble OP and methods that attempt to measure all OP, eg, that associated with surfaces of solid particles?

*Response:*

*The methanol-soluble OP measured in our study cannot be called the total OP measured by Gao et al. (2017). In our method, we sonicated punches of $PM_{2.5}$ filters in methanol, and filtered the suspensions through a 0.45 μm PTFE syringe filter. The filtered extracts were then concentrated to less than 50 μL using a nitrogen dryer to evaporate methanol and were subsequently reconstituted in deionized water (DI). In comparison to this method, Gao et al. (2017) (cited in Table S2) measured $OP^{DTT}$ by three methods. In their first method, they extracted the filters sequentially in water followed by methanol. After sonication, both suspensions were filtered through a 0.45 μm PTFE syringe filter. The subsequent methanol extracts were concentrated to ~200 μL by evaporating methanol, and were then reconstituted in DI. Note, neither ours (direct extraction in methanol followed by filtration) nor their first method (sequential extraction followed by filtration) measure the activity of methanol-insoluble fraction of $PM_{2.5}$ and therefore cannot be termed as total OP.*

*In Gao et al. (2017)'s second method, they directly sonicated punches of $PM_{2.5}$ filters in methanol and removed the filter punches after sonication. The methanol extracts were concentrated to ~200 μL without being filtered, and were then reconstituted in DI. This method could include the activity of water-insoluble and also methanol-insoluble species via surface reaction, but probably not to 100 % efficiency because some of the particles could always remain on the filter fibers irrespective of the solvent used for extraction.*

*In Gao et al. (2017)'s third method, they sonicated filter punches in a mixture of DI and potassium phosphate buffer (K-PB, pH = 7.4) and directly measured OP of the suspensions containing filter punches, without filtering anything. Since this method includes the contribution of even those particles which are not extracted and remain on the filters, we believe that out of all these methods, only this approach can be termed as the total OP. This was further demonstrated from the results of Gao et al. (2017), showing a 5 – 18 % higher average $OP^{DTT}$ obtained from this method compared to earlier two methods.*

*Overall, the "$OP^{DTT}$" obtained from their first method (i.e. the summation of water-soluble OP and the subsequent methanol-soluble OP) was most similar with the methanol-soluble OP measured in our study, but none of them can be considered as the "total OP".*

**Reviewer: Anonymous Reviewer #1**

In this work, the authors measured oxidative potential (OP) of particulate matter in five urban areas in midwestern US. Particulate matter (PM) is a significant health hazard, and its oxidative potential is thought to be representative of its toxicity. The authors assessed oxidative potential in 5 different endpoints on a weekly basis. These OP measurements are often difficult to make, but the authors had developed a system to automate the measurements of PM on filters. The results from the study showed large variabilities across sites and endpoints, and these variabilities, along with poor correlation with PM mass, suggest that PM2.5 mass alone is a poor indicator of potential health impacts. The discussion of the results was not very deep, and, in many cases, more detailed exploration is encouraged to better understand these results. In general, the manuscript is well written, but some of the main messages can be more clearly communicated, rather than buried in a lot of numbers and text. I believe that this manuscript should be published in ACP after some major revisions.

*Response:*

*We thank the reviewer for providing these valuable comments. In the revised manuscript, we have tried our best to reduce the unnecessary information (such as numbers and text) so that the main message of our study become clear. We have also enriched our discussion as well as the conclusion to explicitly state the main take-away message from our exploration. In the following section, we have addressed the reviewer's comments on point-by-point basis.*

Major comments:

In general, this work reads like a measurement report. I was very impressed by the ability to make all these measurements, but somewhat disappointed with the lack of insights from the measurements. More specifically:

-    A lot of information about each site was given in Section 2.1, but when discussing the spatiotemporal variability, there is virtually no discussion in these contexts in Section 3.3. Why does CMP behave so differently? What are the spikes? The same goes for Section 3.5, where the site-to-site comparison is discussed in the context of some statistical measures (correlation coefficient, COD). Again, what are the physical insights?

*Response:*

*This is the first manuscript in the series of papers we plan to write from our yearlong Midwest sampling campaign. In addition to the OP analysis, we are also conducting a lot of chemical and mechanistic analysis (e.g. separation of PM components) on these samples, which we plan to present in our subsequent manuscripts. The current manuscript is expected to serve as the reference for all those subsequent papers and therefore we have to provide as much information as possible about the sampling sites in this manuscript. We understand that all of this information might not be relevant at the current stage given this manuscript is limited to only OP analysis. However, we believe that as our further analysis (i.e. chemical and mechanistic analysis) will emerge, some of this information could become relevant. We further note that the scope of this manuscript was to discuss the patterns of spatiotemporal variability of $PM_{2.5}$ OP in the Midwest US. Therefore, description of the site features in Section 2.1 was intended to justify different classification of the sites, i.e. urban, roadside and rural.*

"Why does CMP behave so differently? What are the spikes?"

*CMP was the only site which was adjacent (< 10 m) to a major urban road (University Avenue in Urbana, IL) and was on the roof of a parking garage, indicating that $PM_{2.5}$ collected at this site was directly impacted by the daily traffic. Our previous study conducted at the same site, Wang et al. (2018) has reported large variations in several redox-active metals, including Cu (4 – 60 ng/m³), Fe (2 – 15 ng/m³), Mn (0.4 – 3 ng/m³), Pb (0.02 – 2.5 ng/m³) and Zn (3 – 10.5 ng/m³), which are all related with the vehicles (both exhaust and non-exhaust emissions). Since the spikes occurring in water-soluble OP at CMP (Figure 3) were generally observed for SLF-based endpoints (i.e. $OP^{AA}$, $OP^{GSH}$ and $OP^{OH-SLF}$), which are all known to be highly sensitive towards metals (Ayres et al., 2008;Calas et al., 2018;Fang et al., 2016;Moreno et al., 2017;Charrier and Anastasio, 2015;Wei et al., 2018), we expect a larger contribution of the variation in daily traffic intensity in the spikes observed at CMP. Note, the $OP^{AA}$ – an endpoint known to be highly sensitive towards Cu (Ayres et al., 2008;Gaetke and Chow, 2003) emitted from brake wear (Hulskotte et al., 2007;Garg et al., 2000;Gietl et al., 2010), showed the most frequent spikes. In comparison to CMP, all other sites were relevantly farther (closest was STL ~230 m) to be directly affected by the road emissions. Thus, such a different behavior of CMP is probably related to its close proximity to a major roadway. We have included this discussion in lines 319 – 327, "A significant temporal variation was observed for CMP with several spikes in the OP activities throughout the year, most prominently for $OP^{AA}$ (Figure 3). These spikes might be attributed to the traffic, as CMP is the only site adjacent (< 10 m) to a major urban road and located on the roof of a parking garage. One of our previous studies, Wang et al. (2018), reported large variations in several redox-active metals (e.g. Cu, Fe, Mn, Pb and Zn), which have been known to be related with the vehicular emissions (Hulskotte et al., 2007;Garg et al., 2000;Gietl et al., 2010;Apeagyei et al., 2011;Councell et al., 2004) at the same CMP site. Since SLF-based endpoints have been shown to be highly sensitive towards metals (Ayres et al., 2008;Calas et al., 2018;Fang et al., 2016;Moreno et al., 2017;Charrier and Anastasio, 2015;Wei et al., 2018), the temporal variation in traffic intensity probably contributes to the spikes observed at CMP. "*

"The same goes for Section 3.5, where the site-to-site comparison is discussed in the context of some statistical measures (correlation coefficient, COD). Again, what are the physical insights? "

*The coefficient of divergence (COD) is a standard measure which has been used in several past studies to explore the spatiotemporal variability in an environmental attribute (Kim et al., 2005;Cheung et al., 2011;Massoud et al., 2011;Verma et al., 2011;Daher et al., 2013;Fang et al., 2014;Huang et al., 2015;Gao et al., 2017;Mukherjee et al., 2019;Feinberg et al., 2019). The primary purpose of Section 3.5 was to compare the COD and correlation coefficient (r) for different OP endpoints versus mass concentration of $PM_{2.5}$. We believe that the key physical insight from section 3.5 (section 3.4 in the revised manuscript) is that there is a larger spatial variability in OP than the $PM_{2.5}$ mass, as revealed from the CODs and r, indicating that the spatial distributions for OP are potentially more affected by the chemical components rather than $PM_{2.5}$ mass. Large variations and weak correlations in most OP endpoints among different sites indicate a more significant effect of the local sources on OP compared to the regional sources. This message has been clearly outlined in lines 518 – 520.*

- Lines 257 to 280 were very hard to follow. The discussion jumped around from OP measure to another (sometimes mass-normalized, other times volume-normalized). The OP endpoints from this particular study were compared to those reported in the literature, but the discussion focuses on very shallow comparisons (e.g. higher, lower, different, the same). I am very confused about the purpose of this discussion: are these comparisons meant to validate the measurements? Are they meant to highlight the differences to illustrate differences between sources, or site characteristics? Are we expecting the OPs to be the same, or different from previous studies? My suggestion is to focus on some main message, and then show the comparisons that illustrate the point.

*Response:*

*We apologize for the reviewer's confusion. However, we differ from the reviewer's point on the discussion jumping from one OP measure to another (sometimes mass-normalized, other times volume-normalized). We are actually following a consistent structure for discussing these five endpoints in the entire manuscript (including this section). SLF-based endpoints were generally discussed first, in the sequence of $OP^{AA}$, $OP^{GSH}$ and $OP^{OH-SLF}$, followed by DTT-based endpoints (first $OP^{DTT}$, and then $OP^{OH-DTT}$). For each endpoint, we first discuss the mass-normalized OP, and then volume-normalized OP. Methanol-soluble OP were discussed after water-soluble OP, following the same sequence as described above. We suggest the reviewer to keep this flow in mind when reading the lines 257 – 280 to avoid any confusion.*

*The reviewer is correct that the primary purpose of this section was to compare our measurements with those reported in the literature. Here, we have compared the OP obtained from our study with OP activities reported from previous literature using the same or similar techniques as ours. In fact, we have further expanded Table S2 (Table S6 in the revised manuscript), by including the methodology of the assays, following the suggestion of another reviewer (#2), who has appreciated this comparison. Since this is the largest dataset on the OP of $PM_{2.5}$ in the Midwest US, and is one of very few studies in US, where all these OP endpoints have been measured on the same set of samples, we think that it is imperative to have a perspective on the general levels of OP in the Midwest US with the rest of the country and the world. Following the reviewer's suggestion, we have clearly expressed the purpose of this section at the beginning of this paragraph in the revised manuscript (lines 360 – 362).*

*From Table S6, we found that the activities of most OP endpoints measured in our study were generally comparable with the previous literature, i.e. in the typical ranges of previously reported OP levels. Regarding the reviewer's point of illustrating the differences between sources, or site characteristics, we don't think it is practical to have it in our manuscript. There are around 20 studies conducted in more than 30 places cited in this section. It is clearly beyond our scope to look into the site characteristics of all these studies and explain our OP results based on that. Moreover, as we have mentioned earlier, we plan to discuss the source apportionment results in our subsequent manuscripts, where we could consider to compare the sources in the Midwest US from other regions, as appropriate. But, we don't think it fits in the scope of the current manuscript.*

- How are we supposed to make sense of the large differences between the various endpoints? They are different measures and operate differently, so they are expected to be different. So, if they are significantly different, then what? The suggestion from the authors is to measure all of them, but then how do we make sense of the different numbers, or trends? A closer examination of what each OP is measuring (and what chemical components are most linked with each measure) would be useful.

*Response:*

*We thank the reviewer. This comment is similar to the 1$^{st}$ comment raised by Reviewer #2. Therefore, we would encourage the reviewer to also read our response to that comment (Pages 1 – 4 of this response document). To the specific points raised by this reviewer, we would like to address them one by one:*

"How are we supposed to make sense of the large differences between the various endpoints? They are different measures and operate differently, so they are expected to be different."

*Yes, these are different measures and operate differently; however, they still come under the umbrella term of "OP" and in the scientific community, they have been often used interchangeably. Therefore, it is logically curious to know if they really produce different results and if so, to what extent, towards the same PM$_{2.5}$. It would be somewhat irrational to assert that without measuring all of them and comparing their outcomes from the same set of PM$_{2.5}$ samples. There have been some studies in the past which have compared their responses on the same set of samples but these are either based on small sample size or have used only few selected assays. A systematic comparison of all these OP assays, particularly in geographical regions of the United States, is lacking and this is the gap our study is trying to fill-in.*

"So, if they are significantly different, then what? The suggestion from the authors is to measure all of them, but then how do we make sense of the different numbers, or trends?"

*This is a good question. From our current investigation, we cannot say which of these assays is the best in terms of representing the health effects. All we know is that the responses of these assays do not correlate with each other. To understand the health relevance of these assays, we first need to integrate them in an epidemiological study, which is beyond the scope of our current study. Some previous studies have adopted this approach for investigating the health relevance of OP by associating it with the health endpoints (Abrams et al., 2017;Strak et al., 2017;Zhang et al.,*

*2016;Yang et al., 2016;Weichenthal et al., 2016a;He et al., 2021;Janssen et al., 2015). These studies have definitely helped in enhancing our understanding on the relevance of OP measurements and the role of specific endpoints in comparison to the PM mass. However, these are very limited with their focus only on 2 or 3 endpoints. Incorporating all the available OP endpoints measured on the same set of samples in epidemiological studies, will help to clearly see their roles and rank them as per their relevance. Therefore, what we mean by "measure all of them" is to develop a database on all these endpoints so that it can be integrated in the epidemiological studies. This will eventually help to evaluate their associations with the health effects and rank them based on their biological relevance. We have modified our discussion on lines 576 – 585 to further clarify our point, "Overall, a poor-to-moderate and inconstant intercorrelation trend among different endpoints of both water-soluble and methanol-soluble OP at most sites indicates that all these assays could be deficient from being ideal and measuring a single endpoint is not enough to represent the overall OP activity. Although the OP endpoints used in our study have covered some of the well-known and important pathways of the in vivo oxidative stress caused by $PM_{2.5}$, there are other endpoints (e.g. consumption of cysteine, formation of $H_2O_2$, etc.), and more assays can be developed in the future. We suggest that a collection of diverse range of OP endpoints, measured separately as done in our study could better capture the role of different PM components and their interactions via different pathways for driving the oxidative levels of the PM in a region. However, it should be noted that our study is not designed to assess and rank the biological relevance of these acellular endpoints, which will require an integration of these and possibly other novel assays involving different routes of oxidative stress, in either toxicological or epidemiological studies."*

- Given that ACP is an chemistry-focused journal, I believe that discussion of chemical composition is well within the scope of this manuscript, and should not be separated for a later publication. Chemical composition is central to many of the questions I posed, and including some information of composition is necessary to make sense of these measurements.

*Response:*

*We partly agree with the reviewer's comment that chemical composition could explain some of the questions raised by the reviewer. However, at the same we want the reviewer to understand that unlike OP, chemical composition is not about making 4 or 5 measurements. We are currently in the process of measuring several chemical species which include EC, OC, WSOC, $NO_3^-$, $SO_4^{-2}$, $NH_4^+$, trace elements (Cu, Fe, Mn, Zn, K, Al, V, Cr, Ni, Sr, Ba, Pb, As and Se), brown carbon, PAHs, hopanes, steranes, alkanes, organic acids and organic nitrogen compounds. Since OP is property inherently linked with the chemical components and their sources, we believe that to properly explain the trends of various OP endpoints, we really need to measure all of these species which have been directly or indirectly linked with the OP. Moreover, before linking the chemical components with OP, we will need to explain their spatiotemporal trends as well. Given current length of the manuscript (18 pages), including all this information will further complicate and convolute the clear message (i.e. the divergent behavior of OP vs. $PM_{2.5}$ mass), it is currently delivering. Again, we agree that chemical composition is important for the OP, but it is not so straight forward. The previous research from our own group (Xiong et al., 2017;Yu et al., 2018) and others (Charrier and Anastasio, 2015;Gonzalez et al., 2017;Lin and Yu, 2020, 2021;Dou et al., 2015) have shown that there are both synergistic and antagonistic interaction among the PM*

*chemical components to alter an OP response. Including some description of the chemical components in the current manuscript might allow us to conduct a shallow analysis of their linkages with the OP, but will prevent us to conduct a thorough analysis in the future manuscript, which we think is more important. Therefore, we believe this should be a separate topic altogether in which we will not only link the OP with the chemical components, but also their interactions as well as their sources, and we plan to address it in our next manuscript. Including all these analysis in the current manuscript, which is focused on exploring the spatiotemporal trends of OP in the Midwest US and its comparison with the PM$_{2.5}$ mass, will unnecessarily lengthen it and mix the important messages we plan to provide through these investigations.*

Minor comments:

- Line 18 and elsewhere: it might useful to define what volume means. Presumably this is air volume, not particle volume

*Response:*

*Yes, the "volume" in "volume-normalized OP" is the volume of sampled air for PM$_{2.5}$ samples analyzed for a particular OP endpoint. We have clarified this term in the revised manuscript in lines 236 – 239, "The mass-normalized (intrinsic, OPm) and volume-normalized (extrinsic, OPv) OP levels were obtained by dividing the blank corrected OP activities by the extracted PM$_{2.5}$ mass (for OPm) and by the volume of air collected on the extracted fractions of filters (for OPv), respectively. The detailed calculations of OPm and OPv have been previously described in Yu et al. (2020)."*

- The introduction is very well-written and reflects the current state of knowledge.

*Response:*

*We thank the reviewer for their comment. We have further enriched the introduction by including more references in lines 64 – 73 , "Calas et al. (2018) compared the responses of several OP endpoints [i.e. OP$^{DTT}$, OP$^{AA}$, OP$^{GSH}$, and electron spin resonance (OP$^{ESR}$)] on PM$_{10}$ samples (N = 98) collected from Chamonix (France). Yang et al. (2014) also used four OP endpoints [OP$^{AA}$, OP$^{DTT}$, OP$^{ESR}$ and reductive acridinium triggering (OP$^{CRAT}$)] to investigate the effect of different extraction solvents and filter types on OP responses using the PM$_{2.5}$ samples (N = 20) collected from two cities (Rotterdam and Amsterdam) in Netherland. The comparison of OP$^{AA}$, OP$^{DTT}$ and OP$^{GSH}$ has been shown in two studies (Fang et al., 2016;Gao et al., 2020), both from the southeast US. We are not aware of any study which has compared ·OH generation in SLF or DTT with other endpoints based on antioxidants consumption (e.g. AA or GSH consumption). Clearly, the studies systematically comparing the responses of these different endpoints on a large sample-set collected at an extensive spatial scale, particularly in the United States are very limited.", and lines 82 – 89, "Globally, the spatiotemporal profiles of OP have been characterized for some geographical regions such as Los Angeles Basin (Saffari et al., 2014, 2013), Denver (Zhang et al., 2008), Atlanta (Fang et al., 2016;Verma et al., 2014) in US, Ontario (Canada) (Jeong et al., 2020;Weichenthal et al., 2019;Weichenthal et al., 2016a),* *France (Borlaza et al., 2021;Calas et al., 2019;Weber et al., 2018;Weber et al., 2021), Italy (Cesari et al., 2019;Perrone et al.,*

*2019;Pietrogrande et al., 2018), Athens in Greece (Paraskevopoulou et al., 2019), Netherland (Yang et al., 2015a;Yang et al., 2015b), and some coastal cities of Bohai [Jinzhou, Tianjin and Yantai (Liu et al., 2018)] and Beijing (Yu et al., 2019;Liu et al., 2014) in China."*

- Lines 85-93: this might be a good place to define some research questions and hypotheses, and address them accordingly at the end. It will help with adding some depth to the discussion and going beyond just reporting measurements.

*Response:*

*We thank the reviewer for their valuable suggestion. We have revised this paragraph to include the research questions of this manuscript and clearly state our hypothesis. The revised paragraphs in lines 100 – 102 read as, "The goal of this analysis is to compare the spatiotemporal distribution of $PM_{2.5}$ OP with that of the mass concentrations. We also want to investigate if different measures of OP, i.e. $OP^{AA}$, $OP^{GSH}$, $OP^{OH-SLF}$, $OP^{DTT}$ and $OP^{OH-DTT}$ show different spatiotemporal trends or are correlated with each other." The research questions raised here are subsequently addressed in different sections (Sections 3.1, 3.2, 3.4 and 3.6) of the manuscript. We have further tried to clarify the main message of our analysis in these sections.*

- Line 100: "Chicago, Indianapolis and St. Louis" seem redundant.

*Response:*

*We have corrected this sentence in the revised manuscript in lines 111 – 113, "while three major city sites [i.e. Chicago (CHI), Indianapolis (IND) and St. Louis (STL)] are representatives of urban background regions of these respective cities."*

- Section 2.2: are the methanol extracts also kept the same PM mass for OP measurement? In the water soluble extract, the volume of water was adjusted to achieve the same mass; how was this done for the methanol soluble extract?

*Response:*

*Yes, the concentrations of $PM_{2.5}$ in the reaction mixtures used for methanol-soluble OP were kept same as those for water-soluble OP measurement (i.e. 50 µg/mL for SLF-based endpoints, and 30 µg/mL for DTT-based endpoints). We first extracted the same area of the filters as that used for the water-soluble OP in 10 mL methanol, and then filtered the extracts through a 0.45 µm PTFE syringe filter. Methanol in the filtered extracts was then evaporated using a nitrogen dryer, and the dried methanol extracts were reconstituted in DI to reach exactly the same volume as the corresponding water-soluble extracts. We have included this detail on lines 176 – 178 of the revised manuscript, "The filtered extracts were then concentrated to less than 50 µL using a nitrogen dryer to evaporate methanol, and were subsequently reconstituted in DI to the exact same volume as the water-soluble extracts."*

- Line 160: when the dried methanol extract was reconstituted in water (DI water), are there insoluble components? For example, I can imagine some organic compounds are extracted by methanol and stick to the walls of the vial when dried, but does not dissolve in water during reconstitution.

*Response:*

*This is a reasonable point. To minimize the bias caused by this deposition loss, we never completely dried the methanol extracts. Rather, we evaporated them to ~50 µL, followed by addition of water to allow the resuspension of the water-insoluble species in water. Moreover, the DI-reconstituted methanol-soluble extracts were always vigorously shaken using an analog vortex mixer (VWR International, Batavia, IL, US) for at least 60 seconds at 3200 rpm to ensure a thorough flush of the organic species which could have been deposited along the wall of the vials. We have revised our manuscript to include these details in lines 176 – 180, "The filtered extracts were then concentrated to less than 50 µL using a nitrogen dryer to evaporate methanol, and were subsequently reconstituted in DI to the exact same volume as the water-soluble extracts. Reconstituted methanol extracts were vigorously shaken on an analog vortex mixer (VWR International, Batavia, IL, US) for at least 60 seconds at 3200 rpm to ensure a thorough flushing of the components probably deposited along the wall of the vials during evaporation."*

- Lines 235-236: 5.7-21.7 does not seem to be significantly higher than 2.0-20.2. Perhaps show the median?

*Response:*

*We thank the reviewer's suggestion. We have included the median of the PM$_{2.5}$ mass concentrations in lines 265 – 268, "Generally, the more urbanized sites of our study (i.e. CHI, STL and IND) showed slightly higher mass concentrations (5.7 – 21.7 µg/m$^3$, median: 11.8 µg/m$^3$) compared to the smaller cities like CMP and its rural component (i.e. BON) (2.0 – 20.2 µg/m$^3$, median: 9.2 µg/m$^3$)". As can be seen, the median at more urbanized sites is slightly higher than the small city sites.*

- Lines 240 and 281: how is the "time series" different from the temporal variation in "spatiotemporal variation"? There are a lot of overlapping points between Sections 3.2 and 3.3, and these sections are be significantly combined and condensed for easier reading. Or perhaps the author intended the discussions to be separate, and if so, it would be good to convey the differences in the section titles.

*Response:*

*Figure 3 and 4 (described in section 3.2) gives a snapshot of the overall trend of OP at all the sites. Although, the time-series plot with all its data points gives an idea of the overall picture, it is unable to clearly illustrate the seasonal and spatial variations, which can be easily masked by the outliers or extreme values. To quantify these variations, we computed the seasonal averages (± standard deviation), which are shown in Figures 5 and 6 (described in section 3.3). However, we agree with the reviewer that both sections are essentially focused on explaining the spatiotemporal variability. Therefore, we combined sections 3.2 and 3.3 in the revised manuscript as "Section 3.2 Spatiotemporal variation in PM$_{2.5}$ OP", and rearranged the paragraphs for a*

*more clarified discussion, while retaining all four figures (i.e. Figures 3-6) for their original purposes.*

- Line 248-249: Just want to confirm: In line 217, the July 4th data were excluded from the regression analysis, but are included here in the discussion. It is a little confusing; perhaps some slight clarification would be useful.

*Response:*

*Yes, the OP data in the week of July 4$^{th}$ were included in the analysis of spatiotemporal variability but excluded from the regression analysis. This is to avoid the potential bias caused by a strong but an episodic event in the regression analysis. We have clarified this in the revised manuscript in lines 247 – 250, "All PM$_{2.5}$ samples were assessed for spatiotemporal variability. However, since several OP endpoints (e.g. OP$^{AA}$, OP$^{GSH}$ and OP$^{DTT}$) were abnormally elevated in the week of July 4th (Independence Day celebration; discussed in section 3.2), we removed this week's sample from our regression analysis to avoid any bias caused by this episodic event."*

- Line 294: why is different from SE US? The seasonal trend seems to be related to photochemical activity (higher in the summer). In general, the midwestern US provides an interesting contrast to previous studies because it has larger temperature differences between summer and winter.

*Response:*

*We thank the reviewer for this interesting observation. We agree that the midwestern US provides an interesting contrast to the previous studies given the larger temperature differences (up to 100 °F) between summer and winter here. This large temperature variation could drive the seasonal variability to some extent. However, it could be that the emission sources in these two seasons (summer vs. winter) are substantially different. For example, Verma et al. (2014) reported highest contributions to OP$^{DTT}$ from biomass burning in winter (47 %) and from secondary organic aerosol in summer (46 %). Higher OP$^{DTT}$ during winter in the Southeast US was attributed to the higher intrinsic redox activity of biomass burning aerosols than those formed during secondary oxidation (Verma et al., 2015). Since we haven't yet done the source apportionment on this dataset, it would be unreasonable to compare the dominant sources (and their seasonality) for OP of our study with Verma et al. (2014). However, we plan to investigate these differences in our subsequent publication.*

- Line 350-355: this seems like a somewhat handwavy explanation for an anomaly, not really supported by evidence. What is the evidence for significant alkaloid compounds at this one particular site? Are there other studies that show Cu can complex with organic compounds and reduce OP?

*Response:*

*We agree that from our study, there is no direct evidence for the high levels of alkaloid compounds at CMP. However, the antagonistic interactions between Cu and certain organic species on OP*

*have been reported in multiple studies. Our previous studies also revealed antagonistic interaction of Cu with quinones, Suwannee River fulvic acid (SRFA) and ambient humic-like substances (HULIS) for both $OP^{DTT}$ and $OP^{OH-DTT}$ (Xiong et al., 2017;Yu et al., 2018). Pietrogrande et al. (2019) also found a suppressing effect of Cu complexing with citric acid on $OP^{AA}$, further substantiating the role of Cu complexes on reducing the OP. In addition to the antagonistic effect of Cu and alkaloid compounds on $OP^{AA}$, Lin and Yu (2020) also found a substantial antagonistic interaction between hydrophilic fraction (which contains high amount of metals) and hydrophobic fraction (mainly organic species) on $OP^{OH-SLF}$. All these studies indicate that the complexation of Cu with organic species has an important role on reducing the OP for various endpoints. Note, the ranges and medians of $M/W^{OP}$ were generally the lowest at CMP for all endpoints (Figure 7), which implies that the complexes of Cu with alkaloid compounds which are efficiently extracted in methanol could probably be responsible for this trend.*

*Considering the reviewer's point that we have not made the specific measurements of these species, we have further toned down our hypothesis based on Cu-complexation with organic compounds in general to explain these results in lines 473 – 479, "The unprotonated nitrogen atom in alkaloids tends to chelate Cu, thus reducing its reactivity with AA. The antagonistic effect of Cu have been reported with other organic compounds (e.g. citric acid) as well (Pietrogrande et al., 2019). Thus, apparently lower levels of methanol-soluble $OP^{AA}$ compared to the water-soluble $OP^{AA}$ at CMP might be associated with the chelation of Cu by these alkaloids or other organic species, which could be more efficiently extracted in methanol."*

- Lines 356-368: why focus on Fe-organic complex? The simpler explanation would be organic compounds that contribute to OP that extracted in methanol but not in water.

*Response:*

*We partially agree with the reviewer that the water-insoluble organic species extracted in methanol could also contribute to the elevated $OP^{OH-SLF}$ and $OP^{OH-DTT}$, however we don't think that this mechanism alone is able to explain the level of elevation observed for these two endpoints (median of $M/W^{OP} = 2.1 – 3.8$ and $1.4 – 1.9$ for $OP^{OH-SLF}$ and $OP^{OH-DTT}$, respectively). Our previous study, Yu et al. (2018) reported moderate activities of $OP^{OH-DTT}$ from multiple types of organic species, including four different quinones, SRFA and ambient HULIS, and nearly zero activity from $Fe^{2+}$ ion. However, much higher activities were observed when mixing $Fe^{2+}$ with all types of organic species (interaction factor, defined as the ratio of the activity of the mixture over the sum of their individual activities = 1.38 – 2.87), indicating the synergistic effect of Fe with organic species for generation ·OH in DTT. Similarly, Gonzalez et al. (2017) and (Wei et al., 2018) also showed a strong synergistic interaction of $Fe^{2+}$ and SRFA through complexation in $OP^{OH-SLF}$. These evidences strongly suggest that complexes of $Fe^{2+}$ with organic compounds have a prominent role in ·OH formation. Wei et al. (2018) also observed that a substantial fraction of Fe gets complexed with hydrophobic organic compounds (28 ± 22 %), which is more efficiently extracted in methanol than water. Moreover, the seasonality of methanol-extracted Fe observed in Wei et al. (2018) followed the same trend as the $M/W^{OP}$ in our study, i.e. the ratio of Fe in 50 % methanol to that in water and $M/W^{OP}$ for $OP^{OH-SLF}v$ in our study were both higher in winter than summer, further suggesting the contribution of Fe-complexes to the increased $OP^{OH-SLF}$ and $OP^{OH-DTT}$ activities of methanol-soluble extracts compared to water-soluble extracts. Therefore, we*

*would like to keep our hypothesis based on Fe-organic complexes to explain these results. However, following the reviewer's suggestion we have also included the possibility of higher OP contributed by the organic compounds extracted in methanol, in lines 482 – 484, "In addition to ·OH-active organic species, e.g. quinones (Charrier and Anastasio, 2015;Xiong et al., 2017;Yu et al., 2018), which are more soluble in methanol, we suspect that one of such components could be organic-complexed Fe."*

- Section 3.6: My suggestion is to point out that current regulations focus on PM mass only, and these results show how inadequate this approach may be. (The reason I suggest this is, at first, I felt it was obvious that OPm would not correlate with PM mass and was somewhat puzzled by the need to do this analysis. But upon second thought, this analysis is useful in a regulatory context.)

*Response:*

*We thank the reviewer for this very important point. We have included it in our discussion in section 3.5 (lines 551 – 552) in the revised manuscript. However, we would like to clarify that we conducted the regression analysis between volume-normalized OP (i.e. OPv and not OPm, which is mass-normalized OP) and PM$_{2.5}$ mass concentrations in Section 3.6. We believe this is what the reviewer meant when they mentioned about the correlation analysis. Since OPm is already normalized by the PM mass, it does not make sense to conduct the correlation between OPm and PM mass. Instead OPv is a property which is in the same equivalent units, i.e. nmol/min/m$^3$ of air as the PM mass (μg/m$^3$ of air), and therefore, they are comparable to perform the regression analysis.*

- Line 474: "the results … provide", not "provides"

*Response:*

*We have corrected this typo in line 616 of the revised manuscript.*

- Figures and tables are generally too complex

*Response:*

*We apologize but we would appreciate if the reviewer could specifically point out which of the figures/tables are complex. We have tried our best to clearly show the information in our figures. All of the figures are either time-series (Figures 2-4), bar charts (Figures 5, 6, 8 and 9) or box-plots (Figure 7), which we believe are very easy to interpret. To make them more legible, we have increased the font sizes of all these figures.*

*Moreover, we have tried to simplify our tables. Specifically, we have combined the average and standard deviation in one column in Table 1, and replaced the P-values with asterisk symbols (\* denotes P < 0.05, \*\* denotes P <0.01) in Tables 3-5.*

**Community: Samuel Weber**

The present study reports the intercomparison of oxidative potential (OP) of PM using different metrics of OP and different extraction protocols. As no consensus has emerged towards which OP method to use, this study is of great interest for documenting various approaches.

However, it should be clarified that it is not the first study of its sort. Namely, Calas et al (2017) have investigated the role of solvent and extraction method and Calas et al (2018) already investigated 5 different OP end-points in Chamonix, France.

Moreover, there is an effort in this manuscript to refer to previous campaign all over the world. We would like to mention to the authors that numerous recent studies in Europe have also reported oxidative potential measurement with multiple assays and have investigated site specificity (Weber et al (2018), Cesari et al (2019), Paraskevopoulou et al (2019), Peronne et al (2019), Pietrogrande et al (2018)), including large-scale variability (Calas et al (2019), Weber et al (2021)) and small-scale variability of OP (Borlaza et al (2021)).

Even if some of the cited studies sampled PM10 and not PM2.5, the discussion of the different OP tests and drivers of OP have been discussed in these papers. These studies should be included in the literature of this manuscript.

Calas, A., Uzu, G., Martins, J. M. F., Voisin, D., Spadini, L., Lacroix, T., and Jaffrezo, J.-L.: The importance of simulated lung fluid (SLF) extractions for a more relevant evaluation of the oxidative potential of particulate matter, Sci Rep, 7, 11617, https://doi.org/10.1038/s41598-017-11979-3, 2017.

Calas, A., Uzu, G., Kelly, F. J., Houdier, S., Martins, J. M. F., Thomas, F., Molton, F., Charron, A., Dunster, C., Oliete, A., Jacob, V., Besombes, J.-L., Chevrier, F., and Jaffrezo, J.-L.: Comparison between five acellular oxidative potential measurement assays performed with detailed chemistry on PM10 samples from the city of Chamonix (France), 18, 7863–7875, https://doi.org/10.5194/acp-18-7863-2018, 2018.

Weber, S., Uzu, G., Calas, A., Chevrier, F., Besombes, J.-L., Charron, A., Salameh, D., Ježek, I., Močnik, G., and Jaffrezo, J.-L.: An apportionment method for the oxidative potential of atmospheric particulate matter sources: application to a one-year study in Chamonix, France, Atmos. Chem. Phys., 18, 9617–9629, https://doi.org/10.5194/acp-18-9617-2018, 2018.

Cesari, D., Merico, E., Grasso, F. M., Decesari, S., Belosi, F., Manarini, F., De Nuntiis, P., Rinaldi, M., Volpi, F., Gambaro, A., Morabito, E., and Contini, D.: Source Apportionment of PM2.5 and of its Oxidative Potential in an Industrial Suburban Site in South Italy, 10, 758, https://doi.org/10.3390/atmos10120758, 2019.

Paraskevopoulou, D., Bougiatioti, A., Stavroulas, I., Fang, T., Lianou, M., Liakakou, E., Gerasopoulos, E., Weber, R. J., Nenes, A., and Mihalopoulos, N.: Yearlong variability of oxidative potential of particulate matter in an urban Mediterranean environment, Atmospheric Environment, 206, 183–196, https://doi.org/10.1016/j.atmosenv.2019.02.027, 2019.

Perrone, M. R., Bertoli, I., Romano, S., Russo, M., Rispoli, G., and Pietrogrande, M. C.: PM2.5 and PM10 oxidative potential at a Central Mediterranean Site: Contrasts between dithiothreitol- and ascorbic acid-measured values in relation with particle size and chemical composition, Atmospheric Environment, 210, 143–155, https://doi.org/10.1016/j.atmosenv.2019.04.047, 2019.

Pietrogrande, M. C., Perrone, M. R., Manarini, F., Romano, S., Udisti, R., and Becagli, S.: PM10 oxidative potential at a Central Mediterranean Site: Association with chemical composition and meteorological parameters, Atmospheric Environment, 188, 97–111, https://doi.org/10.1016/j.atmosenv.2018.06.013, 2018.

Calas, A., Uzu, G., Besombes, J.-L., Martins, J. M. F., Redaelli, M., Weber, S., Charron, A., Albinet, A., Chevrier, F., Brulfert, G., Mesbah, B., Favez, O., and Jaffrezo, J.-L.: Seasonal Variations and Chemical Predictors of Oxidative Potential (OP) of Particulate Matter (PM), for Seven Urban French Sites, 10, 698, https://doi.org/10.3390/atmos10110698, 2019.

Weber, S., Uzu, G., Favez, O., Borlaza, L. J., Calas, A., Salameh, D., Chevrier, F., Allard, J., Besombes, J.-L., Albinet, A., Pontet, S., Mesbah, B., Gille, G., Zhang, S., Pallares, C., Leoz-Garziandia, E., and Jaffrezo, J.-L.: Source apportionment of atmospheric PM10 Oxidative Potential: synthesis of 15 year-round urban datasets in France, 1–38, https://doi.org/10.5194/acp-2021-77, 2021.

Borlaza, L. J. S., Weber, S., Jaffrezo, J.-L., Houdier, S., Slama, R., Rieux, C., Albinet, A., Micallef, S., Trébluchon, C., and Uzu, G.: Disparities in particulate matter (PM10) origins and oxidative potential at a city-scale (Grenoble, France) – Part II: Sources of PM10 oxidative potential using multiple linear regression analysis and the predictive applicability of multilayer perceptron neural network analysis, 1–33, https://doi.org/10.5194/acp-2021-57, 2021.

*Response:*

*We thank Samuel Weber for the useful comments. We agree that it is not the first study to analyze multi-endpoints OP, and there have been studies investigating the spatiotemporal variability and sources of OP using several endpoints. However, all of the studies cited by the reviewer are from Europe. We are not aware of any study which has investigated the spatiotemporal variability of more than 3 OP endpoints in the United States. At most, we could find only two studies both from Southeast US (Atlanta, GA), one of which has compared only two OP endpoints ($OP^{DTT}$ and $OP^{AA}$) (Fang et al., 2016) and another has compared three endpoints ($OP^{DTT}$, $OP^{AA}$ and $OP^{GSH}$) (Gao et al., 2020). Therefore, we have modified our introduction accordingly on lines 63 – 73, "Many of these acellular endpoints have been widely implemented by various researchers for assessing the oxidative properties of PM. Calas et al. (2018) compared the responses of several OP endpoints [i.e. $OP^{DTT}$, $OP^{AA}$, $OP^{GSH}$, and electron spin resonance ($OP^{ESR}$)] on $PM_{10}$ samples (N = 98) collected from Chamonix (France). Yang et al. (2014) also used four OP endpoints [$OP^{AA}$, $OP^{DTT}$, $OP^{ESR}$ and reductive acridinium triggering ($OP^{CRAT}$)] to investigate the effect of different extraction solvents and filter types on OP responses using the $PM_{2.5}$ samples (N = 20) collected from two cities (Rotterdam and Amsterdam) in Netherland. The comparison of $OP^{AA}$, $OP^{DTT}$ and $OP^{GSH}$ has been shown in two studies (Fang et al., 2016; Gao et al., 2020), both from the southeast US. We are not aware of any study which has compared ·OH generation in SLF or DTT with other*

*endpoints based on antioxidants consumption (e.g. AA or GSH consumption). Clearly, the studies systematically comparing the responses of these different endpoints on a large sample-set collected from an extensive spatial scale, particularly in the United States are very limited."*

*We also have included several studies from this list in our manuscript at several appropriate places, e.g. lines 82 – 89 in the introduction, and lines 325 – 327 in the results and discussion section. Table S6 of the manuscript (i.e. Table S2 in the preprint), where we compare our OP levels with other measurements is also updated by including those studies from this list that used the same extraction protocols (i.e. water and methanol extractions as used in our study) and measured OP on PM$_{2.5}$ samples. Inclusion of these studies has enriched our discussion.*

[revised manuscript text omitted]

Out of five samplers used in our study, two were old samplers (about 5 years old, used in various sampling campaigns) and three were brand new, which were bought from TISCH Environmental (Cleves, OH, US) a month before the sampling. These new samplers were factory calibrated and installed at three farther sites, i.e. Chicago (CHI), Indianapolis (IND) and St. Louis (STL). The other two old samplers were installed at Champaign (CMP) and Bondville (BON). For the sole purpose of this discussion, we will name them as CHI (N), IND (N), STL (N), CMP (O) and BON (O). Since the new samplers were factory calibrated, we had more confidence in them, therefore, we chose one of those samplers, i.e. CHI (N), as a reference and compared the responses of other two old samplers, i.e. CMP (O) and BON (O), by running them in pairs, i.e. first CHI (N) and CMP (O) pair, followed by CHI (N) and BON (O) pair, at a site in Urbana in April 2018 (due to some practical constraint, we couldn't run all three of them together). We collected 9 sets of 24-hours integrated Hi-Vol $PM_{2.5}$ samples on quartz filters from each pair, and analyzed them for the DTT assay using the same extraction and analysis procedure as used in our current study. The comparison of $OP^{DTT}$ response was conducted by the orthogonal fit regression analysis of $OP^{DTT}v$ of $PM_{2.5}$ samples collected from CHI (N) and old samplers (**Figure S1**). The correlations between the old samplers and CHI (N) sampler were excellent ($R^2 = 0.92 - 0.94$) with slopes almost equal to 1 ($1.02 - 1.03$), indicating that the samplers collect identical $PM_{2.5}$, and had negligible internal difference in sample collection.

[Figure]

**Figure S1.** Comparison of $OP^{DTT}$ of $PM_{2.5}$ samples collected from CHI (N) sampler with old samplers: (a) CMP (O) sampler; (b) BON (O) sampler.

After the sampling campaign, we again moved the new samplers [i.e. CHI (N), STL (N) and IND (N)] back to CMP site, kept them side-by-side, and collected 9 Hi-Vol samples (24-hours integrated) from each sampler. All these samples were extracted in DI and analyzed for $OP^{DTT}$ in the same manner as used in our current study. The comparison of the reference sampler [i.e. CHI (N)] with other two new samplers was also conducted by orthogonal fit (Figure S2). Excellent correlations ($R^2 = 0.93 - 0.95$) and consistent slopes ($1.05 - 1.06$, close to 1) both showed a good consistency of three new samplers.

[Figure]

**Figure S2.** Comparison of $OP^{DTT}$ of $PM_{2.5}$ samples collected from CHI (N) sampler with other new samplers: (a) STL (N) sampler; (b) IND (N) sampler.

**Table S1.** Dates of samples collection at five sampling sites.

| Season | Week count | Sampling period | CHI | STL | IND | CMP | BON |
|---|---|---|---|---|---|---|---|
| Summer 2018 | 1 | 2018/5/22 – 2018/5/25 | ✓ | ✓ | ✓ | ✓ | ✗ |
| | 2 | 2018/5/29 – 2018/6/1 | ✓ | ✓ | ✓ | ✓ | ✗ |
| | 3 | 2018/6/5– 2018/6/8 | ✓ | ✓ | ✓ | ✓ | ✓ |
| | 4 | 2018/6/12– 2018/6/15 | ✓ | ✓ | ✓ | ✓ | ✓ |
| | 5 | 2018/6/19– 2018/6/22 | ✓ | ✓ | ✓ | ✓ | ✗ |
| | 6 | 2018/6/26– 2018/6/29 | ✓ | ✓ | ✓ | ✓ | ✓ |
| | 7 | 2018/7/3– 2018/7/6 | ✓ | ✓ | ✓ | ✓ | ✓ |
| | 8 | 2018/7/10– 2018/7/13 | ✓ | ✓ | ✓ | ✓ | ✗ |
| | 9 | 2018/7/17– 2018/7/20 | ✓ | ✓ | ✓ | ✓ | ✗ |
| | 10 | 2018/7/24– 2018/7/27 | ✗ | ✓ | ✓ | ✓ | ✓ |
| | 11 | 2018/7/31– 2018/8/3 | ✓ | ✓ | ✓ | ✓ | ✓ |
| | 12 | 2018/8/7– 2018/8/10 | ✓ | ✓ | ✓ | ✓ | ✓ |
| | 13 | 2018/8/14– 2018/8/17 | ✓ | ✓ | ✓ | ✓ | ✓ |
| | 14 | 2018/8/21– 2018/8/24 | ✓ | ✓ | ✓ | ✓ | ✓ |
| | 15 | 2018/8/28– 2018/8/31 | ✓ | ✓ | ✓ | ✓ | ✓ |
| Fall 2018 | 16 | 2018/9/4– 2018/9/7 | ✓ | ✓ | ✓ | ✓ | ✓ |
| | 17 | 2018/9/11– 2018/9/14 | ✓ | ✓ | ✓ | ✓ | ✓ |
| | 18 | 2018/9/18– 2018/9/21 | ✓ | ✓ | ✓ | ✓ | ✓ |
| | 19 | 2018/9/25– 2018/9/28 | ✗ | ✓ | ✓ | ✓ | ✗ |
| | 20 | 2018/10/2– 2018/10/5 | ✗ | ✓ | ✓ | ✓ | ✗ |
| | 21 | 2018/10/9– 2018/10/12 | ✓ | ✗ | ✓ | ✓ | ✓ |
| | 22 | 2018/10/16– 2018/10/19 | ✓ | ✓ | ✓ | ✓ | ✓ |
| | 23 | 2018/10/23– 2018/10/26 | ✓ | ✓ | ✓ | ✓ | ✓ |
| | 24 | 2018/10/30– 2018/11/2 | ✓ | ✓ | ✓ | ✗ | ✓ |
| | 25 | 2018/11/6– 2018/11/9 | ✓ | ✗ | ✓ | ✓ | ✓ |
| | 26 | 2018/11/13– 2018/11/16 | ✓ | ✗ | ✓ | ✓ | ✓ |
| | 27 | 2018/11/20– 2018/11/23 | ✓ | ✓ | ✓ | ✗ | ✓ |
| | 28 | 2018/11/27– 2018/11/30 | ✓ | ✓ | ✓ | ✓ | ✓ |
| Winter 2018 | 29 | 2018/12/4– 2018/12/7 | ✓ | ✓ | ✓ | ✓ | ✓ |
| | 30 | 2018/12/11– 2018/12/14 | ✗ | ✓ | ✓ | ✓ | ✓ |
| | 31 | 2018/12/18– 2018/12/21 | ✗ | ✓ | ✓ | ✓ | ✓ |
| | 32 | 2018/12/25– 2018/12/28 | ✗ | ✓ | ✓ | ✓ | ✓ |
| | 33 | 2019/1/1– 2019/1/4 | ✗ | ✓ | ✓ | ✓ | ✓ |
| | 34 | 2019/1/8– 2019/1/11 | ✗ | ✓ | ✓ | ✓ | ✓ |
| | 35 | 2019/1/15– 2019/1/18 | ✗ | ✓ | ✓ | ✓ | ✗ |
| | 36 | 2019/1/22– 2019/1/25 | ✓ | ✓ | ✓ | ✓ | ✗ |
| | 37 | 2019/1/29– 2019/2/1 | ✓ | ✓ | ✓ | ✓ | ✓ |
| | 38 | 2019/2/5– 2019/2/8 | ✓ | ✓ | ✓ | ✓ | ✓ |
| | 39 | 2019/2/12– 2019/2/15 | ✓ | ✓ | ✓ | ✓ | ✓ |
| | 40 | 2019/2/19– 2019/2/22 | ✓ | ✓ | ✓ | ✓ | ✓ |
| | 41 | 2019/2/26– 2019/3/1 | ✓ | ✓ | ✓ | ✓ | ✓ |
| Spring 2019 | 42 | 2019/3/5– 2019/3/8 | ✓ | ✓ | ✓ | ✓ | ✓ |
| | 43 | 2019/3/12– 2019/3/15 | ✗ | ✓ | ✓ | ✓ | ✓ |
| | 44 | 2019/3/19– 2019/3/22 | ✓ | ✓ | ✓ | ✓ | ✓ |
| | 45 | 2019/3/26– 2019/3/29 | ✓ | ✓ | ✓ | ✓ | ✓ |
| | 46 | 2019/4/2– 2019/4/5 | ✓ | ✓ | ✓ | ✓ | ✓ |
| | 47 | 2019/4/9– 2019/4/12 | ✓ | ✓ | ✓ | ✓ | ✓ |
| | 48 | 2019/4/16– 2019/4/19 | ✓ | ✓ | ✓ | ✗ | ✓ |
| | 49 | 2019/4/23– 2019/4/26 | ✓ | ✓ | ✓ | ✓ | ✓ |
| | 50 | 2019/4/30– 2019/5/3 | ✓ | ✗ | ✓ | ✓ | ✓ |
| | 51 | 2019/5/7– 2019/5/10 | ✓ | ✓ | ✓ | ✓ | ✓ |
| | 52 | 2019/5/14– 2019/5/17 | ✓ | ✗ | ✓ | ✓ | ✓ |
| | 53 | 2019/5/21– 2019/5/24 | ✓ | ✗ | ✓ | ✓ | ✓ |
| | 54 | 2019/5/28– 2019/5/31 | ✓ | ✗ | ✓ | ✓ | ✓ |

The symbol ✓ denotes the collection of a sample, and the symbol ✗ denotes no collection of the sample in that week (due to several reasons such as unfavorable weather conditions, broken sampler, etc.).

**Table S2.** Precision of SAMERA for methanol-soluble OP measurements compared with water-soluble OP measurements.

| Endpoint | Unit | Average | Standard Deviation | CoV (%) | CoV (%) for the water-soluble $PM_{2.5}$ extract (Yu et al., 2020) |
|---|---|---|---|---|---|
| $OP^{AA}$ | nmol/min/m$^3$ | 0.132 | 0.018 | 13.51 | 11.87 |
| $OP^{GSH}$ | nmol/min/m$^3$ | 0.098 | 0.010 | 10.65 | 7.89 |
| $OP^{OH-SLF}$ | pmol/min/m$^3$ | 0.740 | 0.011 | 14.49 | 10.56 |
| $OP^{DTT}$ | nmol/min/m$^3$ | 0.187 | 0.017 | 8.89 | 10.52 |
| $OP^{OH-DTT}$ | pmol/min/m$^3$ | 0.216 | 0.023 | 10.88 | 13.28 |

**Table S3.** Results of 1-way ANOVA test for assessing the temporal and spatial variability of $PM_{2.5}$ mass concentrations.

| Variability | Sampling Site/Season | F value | Significantly different group(s) |
|---|---|---|---|
| Temporal | CHI | 1.95 | |
| | STL | 1.79 | |
| | IND | 0.33 | |
| | CMP | 3.25* | Fall 2018 |
| | BON | 0.82 | |
| Spatial | Summer 2018 | 3.48* | STL |
| | Fall 2018 | 3.13* | CHI, STL, IND, CMP |
| | Winter 2018 | 5.01** | CHI |
| | Spring 2019 | 3.35* | BON |

[revised manuscript text omitted]
 | 483 | Ambient $PM_{2.5}$ samples were collected using a Hi-Vol sampler on quartz filters, extracted in DI and filtered through a syringe filter. $OP^{AA}$ of filtered extracts was assessed with an AA-only assay (no other antioxidants involved; concentration of AA was 200 µM) with an automated system. AA was measured based on a photometric method (at 265 nm). |
| Mudway et al. (2005) | ≤ 2.5 | 0.012 ± 0.0001 nmol·min$^{-1}$·µg$^{-1}$ | Eksaal, India | Biomass burning | 3 | Biomass burning samples were collected from dung-cake combustion, and extracted in Chelex-treated DI with 5% methanol. $OP^{AA}$ of filtered extracts was assessed in a respiratory tract lining fluid (RTLF; composition was 200 µM AA, 200 µM GSH and 200 µM UA). AA was measured based on a photometric method (at 265 nm). |
| Künzli et al. (2006) | ≤ 2.5 | 0.0096 ± 0.0025 nmol·min$^{-1}$·µg$^{-1}$ | 19 European cities | Urban | 716 | Ambient $PM_{2.5}$ samples were collected using a Basel-Sampler, and extracted in metal-free DI. $OP^{AA}$ was assessed in the same manner as Mudway et al. (2005). |
| Szigeti et al. (2016) | ≤ 2.5 | 0.0017 – 0.04 nmol·min$^{-1}$·µg$^{-1}$ | 8 European cities | Urban | 22 | Ambient and indoor $PM_{2.5}$ samples were collected using a Low-Vol sampler, and directly incubated in RTLF having same composition as in Mudway et al. (2005). AA was measured based on a photometric method (at 265 nm). |
| Godri et al. (2011) | 1.0 – 1.9 | 0.0058 ± 0.0025 nmol·min$^{-1}$·µg$^{-1}$ | London, United Kingdom | Urban | 14 | Ambient size-segregated samples were collected using a MOUDI sampler, and extracted in Chelex-treated DI with 5% methanol. $OP^{AA}$ was assessed in the same manner as Mudway et al. (2005). |

| Reference | PM size | OP$^{AA}$ | Location | Type | n | Notes |
|---|---|---|---|---|---|---|
| Perrone et al. (2019) | ≤ 2.5 | $0.006 \pm 0.001$ nmol·min$^{-1}$·µg$^{-1}$
$0.136 \pm 0.020$ nmol·min$^{-1}$·m$^{-3}$ | Lecce, Italy | Urban | 39 | Ambient PM$_{2.5}$ samples were collected using a low volume HYDRA-FAI dual sampler, and extracted in DI. OP$^{AA}$ of filtered extracts was assessed with an AA-only assay similar as in Fang et al. (2016). |
| Gao et al. (2020a) | ≤ 2.5 | $0.023 - 0.126$ nmol·min$^{-1}$·m$^{-3}$ | Atlanta, GA | Urban | 349 | Ambient PM$_{2.5}$ samples were collected using a Hi-Vol sampler on quartz filters, extracted in DI and filtered through a syringe filter. OP$^{AA}$ was assessed in the same manner as Mudway et al. (2005). |
| Yang et al. (2014) | ≤ 2.5 | $0.8 - 35.0$ nmol·s$^{-1}$·m$^{-3}$ | Rotterdam and Amsterdam, Netherland | Urban | 10 | Ambient PM$_{2.5}$ samples were collected using a Harvard Impactor and extracted in ultrapure water. OP$^{AA}$ of filtered extracts was assessed AA-only assay similar as in Fang et al. (2016). |
| Yu et al. (2020) | ≤ 2.5 | $0.004 - 0.077$ nmol·min$^{-1}$·µg$^{-1}$
median: $0.012$ nmol·min$^{-1}$·µg$^{-1}$
$0.044 - 0.745$ nmol·min$^{-1}$·m$^{-3}$
median: $0.160$ nmol·min$^{-1}$·m$^{-3}$ | Midwest US (5 sites) | Urban (4), rural (1) | 54 | PM$_{2.5}$ sampling, preparation and OP$^{AA}$ measurement were conducted in the same manner as the current study. |
| Yang et al. (2014)* | ≤ 2.5 | $2.2 - 43.5$ nmol·s$^{-1}$·m$^{-3}$ | Rotterdam and Amsterdam, Netherland | Urban | 20 | Ambient PM$_{2.5}$ samples were collected using a Harvard Impactor and extracted in methanol. Filtered methanol extracts were evaporated using an evaporator set, and reconstituted with DI. OP$^{AA}$ of water-reconstituted methanol extracts was assessed AA-only assay similar as in Fang et al. (2016). |
| This study | ≤ 2.5 | $0.002 - 0.077$ nmol·min$^{-1}$·µg$^{-1}$
median: $0.007$ nmol·min$^{-1}$·µg$^{-1}$
$0.012 - 0.908$ nmol·min$^{-1}$·m$^{-3}$
median: $0.078$ nmol·min$^{-1}$·m$^{-3}$ | Midwest US (5 sites) | Urban (4), rural (1) | 241 | See section 2 (experimental methods). |
| This study* | | $0.004 - 0.029$ nmol·min$^{-1}$·µg$^{-1}$
median: $0.012$ nmol·min$^{-1}$·µg$^{-1}$
$0.030 - 0.311$ nmol·min$^{-1}$·m$^{-3}$
median: $0.134$ nmol·min$^{-1}$·m$^{-3}$ | Midwest US (5 sites) | Urban (4), rural (1) | 241 | |

Asterisk - * indicates that the reported results are methanol-soluble OP$^{AA}$.

(b) OP$^{GSH}$

| Reference | PM size (μm) | Levels | Location | Location type | Sample size | Methodology |
|---|---|---|---|---|---|---|
| Mudway et al. (2005) | ≤ 2.5 | 0.0083 ± 0.0002 nmol·min$^{-1}$·μg$^{-1}$ | Eksaal, India | Biomass burning | 3 | OP$^{GSH}$ of filtered extracts was measured in RTLF. GSH was measured with a glutathione disulfide (GSSG)-reductase-5,5-dithio-bis-(2-nitrobenzoic acid) (DTNB) recycling assay, based on a photometric method (at 405 nm). |
| Künzli et al. (2006) | ≤ 2.5 | 0.0041 ± 0.0017 nmol·min$^{-1}$·μg$^{-1}$ | 19 European cities | Urban | 716 | OP$^{GSH}$ was assessed in the same manner as Mudway et al. (2005). |
| Szigeti et al. (2016) | ≤ 2.5 | 0 – 0.0275 nmol·min$^{-1}$·μg$^{-1}$ | 8 European cities | Urban | 22 | Punches of filter samples were directly incubated in RTLF, and measured for OP$^{GSH}$ in the same manner with Mudway et al. (2005). |
| Godri et al. (2011) | 1.0 – 1.9 | 0.0042 ± 0.0033 nmol·min$^{-1}$·μg$^{-1}$ | London, United Kingdom | Urban | 14 | OP$^{GSH}$ was assessed in the same manner as Mudway et al. (2005). |
| Gao et al. (2020a) | ≤ 2.5 | 0.025 – 0.067 nmol·min$^{-1}$·m$^{-3}$ | Atlanta, GA | Urban | 349 | OP$^{GSH}$ was assessed in the same manner as Mudway et al. (2005). |
| Yu et al. (2020) | ≤ 2.5 | 0.001 – 0.040 nmol·min$^{-1}$·μg$^{-1}$ median: 0.010 nmol·min$^{-1}$·μg$^{-1}$ 0.008 – 0.463 nmol·min$^{-1}$·m$^{-3}$ median: 0.100 nmol·min$^{-1}$·m$^{-3}$ | Midwest US (5 sites) | Urban (4), rural (1) | 54 | PM$_{2.5}$ sampling, preparation and OP$^{GSH}$ measurement were conducted in the same manner as the current study. |
| This study | ≤ 2.5 | 0.002 – 0.035 nmol·min$^{-1}$·μg$^{-1}$ median: 0.007 nmol·min$^{-1}$·μg$^{-1}$ 0.013 – 0.419 nmol·min$^{-1}$·m$^{-3}$ median: 0.074 nmol·min$^{-1}$·m$^{-3}$ | Midwest US (5 sites) | Urban (4), rural (1) | 241 | See section 2 (experimental methods). |

(c) OP$^{OH-SLF}$

| Reference | PM size (µm) | Levels | Location | Location type | Sample size | Methodology |
|---|---|---|---|---|---|---|
| Vidrio et al. (2009) | ≤ 2.5 | 0.253 ± 0.135 pmol·min$^{-1}$·µg$^{-1}$ | Davis, CA | Urban | ~90 | Ambient PM$_{2.5}$ samples were collected using IMPROVE Version II samplers on Teflo filters, directly incubated in SLF (composition was 114 mM NaCl, 10 mM sodium benzoate, 10 mM total phosphate to buffer the solution at pH 7.4, 200 µM AA and 300 µM CA) with desferoxamine (DSF) for 24 hours, and measured for ·OH generation. ·OH was captured by sodium benzoate and measured based on a photometric method (at 256 nm) using a high-performance liquid chromatography (HPLC). |
| Ma et al. (2015) | ≤ 2.5 | 0.092 ± 0.019 pmol·min$^{-1}$·µg$^{-1}$ | Guangzhou, China | Urban | 72 | Ambient PM$_{2.5}$ samples were collected using a Low-Vol sampler on Teflon filters. OP$^{OH-SLF}$ was measured in the same manner as in Vidrio et al. (2009). |
| Yu et al. (2020) | ≤ 2.5 | 0.085 – 0.967 pmol·min$^{-1}$·µg$^{-1}$ median: 0.307 pmol·min$^{-1}$·µg$^{-1}$ 0.857 – 7.884 pmol·min$^{-1}$·m$^{-3}$ median: 3.559 pmol·min$^{-1}$·m$^{-3}$ | Midwest US (5 sites) | Urban (4), rural (1) | 54 | PM$_{2.5}$ sampling, preparation and OP$^{OH-SLF}$ measurement were conducted in the same manner as the current study. |
| This study | ≤ 2.5 | 0.040 – 1.217 pmol·min$^{-1}$·µg$^{-1}$ median: 0.142 pmol·min$^{-1}$·µg$^{-1}$ 0.269 – 12.13 pmol·min$^{-1}$·m$^{-3}$ median: 1.449 pmol·min$^{-1}$·m$^{-3}$ | Midwest US (5 sites) | Urban (4), rural (1) | 241 | See section 2 (experimental methods). |

(d) OP$^{DTT}$

| Reference | PM size (μm) | Levels | Location | Location type | Sample size | Methodology |
|---|---|---|---|---|---|---|
| Fang et al. (2015) | ≤ 2.5 | 0.010 – 0.097 nmol·min$^{-1}$·μg$^{-1}$ median: 0.024 – 0.041 nmol·min$^{-1}$·μg$^{-1}$ 0.05 – 0.81 nmol·min$^{-1}$·m$^{-3}$ median: 0.23 – 0.31 nmol·min$^{-1}$·m$^{-3}$ | Southeast US | Urban and rural | 503 | Ambient PM$_{2.5}$ samples were collected using a Hi-Vol sampler on quartz filters, extracted in DI and filtered through a syringe filter. Filtered extracts were then incubated in a mixture of 100 μM DTT and 0.5 mM potassium phosphate buffer (K-PB; pH = 7.4). DTT was captured by DTNB and measured based on a photometric method (at 412 nm) using an automated system. |
| Xiong et al. (2017) | ≤ 2.5 | 0.1 – 0.18 nmol·min$^{-1}$·m$^{-3}$ | Urbana, IL | Urban | 10 | Ambient PM$_{2.5}$ samples were collected with Hi-Vol sampler on quartz filters, extracted in Milli-Q water, and filtered through a syringe filter. OP$^{DTT}$ were assessed in the same manner with Fang et al. (2015). |
| Cho et al. (2005) | ≤ 2.5 | 0.013 – 0.047 nmol·min$^{-1}$·μg$^{-1}$ median: 0.029 nmol·min$^{-1}$·μg$^{-1}$ | Los Angeles basin, CA | Urban | 11 | Ambient size-segregated samples were collected using a VACES in conjunction with a BioSampler. Collected suspensions were then incubated in a mixture of 100 μM DTT and 0.5 mM potassium phosphate buffer (K-PB; pH = 7.4). DTT was captured by DTNB and measured based on a photometric method (at 412 nm) at designated time points within 90 min. |
| Charrier and Anastasio (2012) | ≤ 2.5 | 0.02 – 0.061 nmol·min$^{-1}$·μg$^{-1}$ median: 0.029 nmol·min$^{-1}$·μg$^{-1}$ | San Joaquin, CA | Urban, rural | 6 | Ambient PM$_{2.5}$ samples were collected on Teflon filters, but the filter extraction method was not reported. DTT assay was conducted by incubating the aqueous sample extracts in 100 μM DTT. DTT was captured by DTNB and measured based on a photometric method (at 412 nm) at four time points within 16 min. |
| Gao et al. (2017) | ≤ 2.5 | 0.09 – 0.30 nmol·min$^{-1}$·m$^{-3}$ median: 0.19 nmol·min$^{-1}$·m$^{-3}$ | Atlanta, GA (2 sites) | Urban | 66 | PM$_{2.5}$ sampling, preparation and OP$^{DTT}$ measurement were conducted in the same manner as |

| | | | | | | |
|---|---|---|---|---|---|---|
| Gao et al. (2020a) and Gao et al. (2020b) | ≤ 2.5 | 0.005 – 0.070 nmol·min$^{-1}$·µg$^{-1}$ average: 0.024 nmol·min$^{-1}$·µg$^{-1}$ 0.05 – 0.48 nmol·min$^{-1}$·m$^{-3}$ average: 0.22 nmol·min$^{-1}$·m$^{-3}$ | Atlanta, GA | Urban | 349 | PM$_{2.5}$ sampling, preparation and OP$^{DTT}$ measurement were conducted in the same manner as Fang et al. (2015). |
| Hu et al. (2008) | 0.25 – 2.5 | 0.014 – 0.024 nmol·min$^{-1}$·µg$^{-1}$ median: 0.019 nmol·min$^{-1}$·µg$^{-1}$ 0.10 – 0.16 nmol·min$^{-1}$·m$^{-3}$ median: 0.14 nmol·min$^{-1}$·m$^{-3}$ | Los Angeles harbor, CA | Urban | 6 | Ambient size-segregated samples were collected with Sioutas samplers on Zefluor and Quartz filters, and extracted in Milli-Q water. DTT assay was conducted by incubating the PM suspensions in 100 µM DTT at pH = 7.4 adjusted by K-PB. DTT was captured by DTNB and measured based on a photometric method (at 412 nm) at designated time points within 30 min. |
| Cesari et al. (2019) | ≤ 2.5 | 0.012 ± 0.008 nmol·min$^{-1}$·µg$^{-1}$ 0.19 ± 0.10 nmol·min$^{-1}$·m$^{-3}$ | Sarno, Italy | Urban | ~50 | Ambient PM$_{2.5}$ samples were collected using a Low-Vol sequential sampler on quartz filters, extracted in DI and filtered through a syringe filter. DTT assay was conducted by incubating the extracts in DTT (concentration not reported) at pH = 7.4 adjusted by K-PB. DTT was captured by DTNB and measured based on a photometric method (at 412 nm) at designated time points (details not reported). |
| Paraskevopoulou et al. (2019) | ≤ 2.5 | 0.028 ± 0.014 nmol·min$^{-1}$·µg$^{-1}$ 0.33 ± 0.20 nmol·min$^{-1}$·m$^{-3}$ | Athens, Greece | Urban | 361 | Ambient PM$_{2.5}$ samples were collected using a Dichotomous Partisol sampler on quartz filters, extracted in DI and filtered through a syringe filter. OP$^{DTT}$ was assessed in the same manner as Fang et al. (2015). |
| Perrone et al. (2019) | ≤ 2.5 | 0.010 ± 0.001 nmol·min$^{-1}$·µg$^{-1}$ 0.228 ± 0.024 nmol·min$^{-1}$·m$^{-3}$ | Lecce, Italy | Urban | 39 | Ambient PM$_{2.5}$ samples were collected using a low volume HYDRA-FAI dual sampler, and extracted in DI. DTT assay was conducted by incubating the aqueous sample extracts in 100 µM DTT. DTT was captured by DTNB and measured based on a photometric method (at 412 nm) at five time points within 40 min. |

| Yang et al. (2014) | ≤ 2.5 | 0.4 – 7.2 nmol·s$^{-1}$·m$^{-3}$ | Rotterdam and Amsterdam, Netherland | Urban | 10 | Ambient PM$_{2.5}$ samples were collected using a Harvard Impactor and extracted in ultrapure water. OP$^{DTT}$ of water-soluble extracts was assessed in the same manner as Hu et al. (2008). |
|---|---|---|---|---|---|---|
| Yu et al. (2020) | ≤ 2.5 | 0.004 – 0.193 nmol·min$^{-1}$·µg$^{-1}$ median: 0.014 nmol·min$^{-1}$·µg$^{-1}$ 0.041 – 1.282 nmol·min$^{-1}$·m$^{-3}$ median: 0.146 nmol·min$^{-1}$·m$^{-3}$ | Midwest US (5 sites) | Urban (4), rural (1) | 54 | PM$_{2.5}$ sampling, preparation and OP$^{DTT}$ measurement were conducted in the same manner as the current study. |
| Verma et al. (2012)* | ≤ 2.5 | 0.020 – 0.054 nmol·min$^{-1}$·µg$^{-1}$ median: 0.034 nmol·min$^{-1}$·µg$^{-1}$ | Atlanta, GA | Urban | 8 | Ambient PM$_{2.5}$ samples were collected using a Hi-Vol sampler on quartz filters, extracted in both methanol and water, and filtered through a syringe filter. Methanol extracts were evaporated to nearly dryness using a rotary evaporator and reconstituted to 15 mL with 0.1 M K-PB (pH = 7.4). Reconstituted methanol extracts were incubated in 100 µM DTT and 0.5 M K-PB (pH = 7.4). DTT was captured by DTNB and measured based on a photometric method (at 412 nm) at seven time points within 20 min. |
| Gao et al. (2017)* | ≤ 2.5 | 0.14 – 0.47 nmol·min$^{-1}$·m$^{-3}$ median: 0.30 nmol·min$^{-1}$·m$^{-3}$ | Atlanta, GA (2 sites) | Urban | 66 | Method 1: Ambient PM$_{2.5}$ samples were extracted in a stepwise manner with DI and methanol. Both extracts were filtered through a syringe filter. Methanol extracts were evaporated to ~200 µL using high-purity nitrogen and reconstituted with DI. Total OP was calculated by adding the OP of both extracts. Method 2: Samples were extracted in methanol. Punches were removed after sonication. The remaining suspensions were analyzed for OP$^{DTT}$ without being filtered through a syringe filter. Method 3: Samples were sonicated in K-PB (pH = 7.4). The mixture was analyzed for OP$^{DTT}$ without removing inside punches or being filtered through a syringe filter. OP$^{DTT}$ measurement was conducted in the same |

| Study | PM size | OP$^{DTT}$ | Location | Type | N | Notes |
|---|---|---|---|---|---|---|
| | | | | | | manner as Fang et al. (2015) using a modified automated system for analyzing suspensions with insoluble fractions. |
| Gao et al. (2020b)* | ≤ 2.5 | 0.012 – 0.116 nmol·min$^{-1}$·μg$^{-1}$ average: 0.027 nmol·min$^{-1}$·μg$^{-1}$ 0.13 – 0.58 nmol·min$^{-1}$·m$^{-3}$ average: 0.28 nmol·min$^{-1}$·m$^{-3}$ | Atlanta, GA | Urban | 349 | PM$_{2.5}$ sampling, preparation and OP$^{DTT}$ measurement were conducted in the same manner as Gao et al. (2017) (Method 3). |
| Yang et al. (2014)* | ≤ 2.5 | 0.5 – 5.2 nmol·min$^{-1}$·m$^{-3}$ | Rotterdam and Amsterdam, Netherland | Urban | 20 | Ambient PM$_{2.5}$ samples were collected using a Harvard Impactor and extracted in methanol. Filtered methanol extracts were evaporated using an evaporator set, and reconstituted with DI. OP$^{DTT}$ of water-reconstituted methanol-soluble extracts was assessed in the same manner as Hu et al. (2008). |
| This study | ≤ 2.5 | 0.004 – 0.032 nmol·min$^{-1}$·μg$^{-1}$ median: 0.014 nmol·min$^{-1}$·μg$^{-1}$ 0.029 – 0.561 nmol·min$^{-1}$·m$^{-3}$ median: 0.150 nmol·min$^{-1}$·m$^{-3}$ | Midwest US (5 sites) | Urban (4), rural (1) | 241 | See section 2 (experimental methods). |
| This study* | ≤ 2.5 | 0.004 – 0.042 nmol·min$^{-1}$·μg$^{-1}$ median: 0.021 nmol·min$^{-1}$·μg$^{-1}$ 0.031 – 0.639 nmol·min$^{-1}$·m$^{-3}$ median: 0.234 nmol·min$^{-1}$·m$^{-3}$ | Midwest US (5 sites) | Urban (4), rural (1) | 241 | |

Asterisk - * indicates that the reported results are methanol-soluble OP$^{DTT}$.

(e) OP$^{OH-DTT}$

| Reference | PM size (µm) | Levels | Location | Location type | Sample size | Methodology |
|---|---|---|---|---|---|---|
| Xiong et al. (2017) | $\leq 2.5$ | $0.2 - 0.6$ pmol·min$^{-1}$·m$^{-3}$ | Urbana, IL | Urban | 10 | PM$_{2.5}$ extracts were incubated in 100 µM DTT and K-PB (pH = 7.4) with 50 mM TPT. ·OH was captured by TPT and measured based on a fluorometric method (excitation/emission wavelength of 310/425 nm) at six time points within 120 min. |
| Yu et al. (2018) | $\leq 2.5$ | $0.2 - 1.1$ pmol·min$^{-1}$·m$^{-3}$ | Urbana, IL | Urban | 10 | PM$_{2.5}$ sampling, preparation and OP$^{OH-DTT}$ measurement were conducted in the same manner as Xiong et al. (2017). |
| Yu et al. (2020) | $\leq 2.5$ | $0.034 - 0.357$ pmol·min$^{-1}$·µg$^{-1}$ median: 0.082 pmol·min$^{-1}$·µg$^{-1}$ $0.360 - 4.152$ pmol·min$^{-1}$·m$^{-3}$ median: 1.054 pmol·min$^{-1}$·m$^{-3}$ | Midwest US (5 sites) | Urban (4), rural (1) | 54 | PM$_{2.5}$ sampling, preparation and OP$^{OH-DTT}$ measurement was conducted in the same manner as the current study. |
| This study | $\leq 2.5$ | $0.004 - 0.357$ pmol·min$^{-1}$·µg$^{-1}$ median: 0.065 pmol·min$^{-1}$·µg$^{-1}$ $0.022 - 3.565$ pmol·min$^{-1}$·m$^{-3}$ median: 0.722 pmol·min$^{-1}$·m$^{-3}$ | Midwest US (5 sites) | Urban (4), rural (1) | 241 | See section 2 (experimental methods). |

[revised manuscript text omitted]